# Transferable Modulation of Cognitive Control: The Cross-Task Role of Conflict Adaptation in Thematic Roles Assignment in Chinese

**DOI:** 10.3390/bs15070899

**Published:** 2025-07-02

**Authors:** Jiefei Luo, Qi Cheng, Mengfang Zhang, Yan Wu

**Affiliations:** 1School of Psychology, Northeast Normal University, Changchun 130024, China; luojf057@nenu.edu.cn (J.L.); mfzhang901@nenu.edu.cn (M.Z.); wuy399@nenu.edu.cn (Y.W.); 2Kindergarten Affiliated of Northeast Normal University, Northeast Normal University, Changchun 130022, China

**Keywords:** cognitive control, conflict adaptation, cross-task transfer, thematic role assignment, sentence comprehension

## Abstract

Conflict adaptation reflects the dynamic modulation of information processing by the cognitive control system following conflict detection. A central question in language processing research concerns whether control elicited by non-linguistic tasks generalizes across tasks to influence higher-order processes such as sentence comprehension. The present study employed color-word Stroop tasks of varying complexity and, in conjunction with eye-tracking technology, examined their cross-task regulatory effects of conflict adaptation on thematic role assignment in Chinese. Across two experiments, participants read sentences containing either congruent or conflicting thematic roles following Stroop trials with congruent or incongruent stimuli. The temporal dynamics of syntactic processing were captured via eye movement measures. Results indicated that both conflict tasks triggered cross-task conflict adaptation, as evidenced by accelerated syntactic processing and reduced regression behaviors when thematically incongruent sentences followed conflict trials. Notably, the more complex color-word Stroop task imposed greater demands on cognitive control resources and elicited earlier cognitive adaptation effects during comprehension. Theoretically, these findings extend conflict monitoring theory to the domain of language processing, demonstrating that cognitive control mechanisms contribute to real-time syntactic parsing. Methodologically, the use of eye-tracking to examine thematic role assignment provides fine-grained empirical evidence for the interaction between domain-general control and language-specific processing.

## 1. Introduction

Cognitive conflict refers to the interference of task-irrelevant information with task-relevant information, typically arising when individuals are required to make decisions among competing options or information representations during the course of information processing ([10]; [51]). Empirical evidence suggests that encountering cognitive conflict activates the cognitive control system ([53]), leading to dynamic adjustments in information processing that culminate in the emergence of conflict adaptation. Conflict adaptation is characterized by improved processing efficiency following the experience of cognitive conflict, thereby showing better control and processing abilities in subsequent similar tasks ([33]). However, existing research on conflict adaptation has predominantly concentrated on performance within classic cognitive control paradigms, such as the Flanker task (e.g., [12]; [33]; [72]; [76]) or the Stroop task (e.g., [42]; [67]; [71]). Whether conflict adaptation can extend across task boundaries—particularly from general cognitive control tasks (e.g., the Stroop task) to conflict resolution in reading comprehension—remains an open question warranting further empirical investigation. Accordingly, the present study is designed to address two primary objectives: first, to determine whether conflict adaptation effects can generalize across distinct task domains; and second, to explore the role of cognitive control mechanisms in reading comprehension. Through this dual focus, the current study aims to offer novel empirical insights into the cross-domain interplay between cognitive control and language processing.

Evidence supporting the conflict adaptation effect has largely been derived from color-word Stroop paradigms that elicit conflict through systematic manipulation of stimulus congruency. A prototypical example is the classical color-word Stroop task employed by [42] ([42]), which required participants to respond to the ink color of color words. This paradigm included two conditions: a congruent condition (e.g., the word “red” printed in red ink) and an incongruent condition (e.g., the word “red” printed in green ink). The study demonstrated that activation intensity in the anterior cingulate cortex (ACC) during incongruent trials significantly predicted subsequent changes in both neural activity and behavioral adjustments in subsequent conflict trials. Behaviorally, participants exhibited significantly reduced reaction times in subsequent incongruent trials, with performance in some instances approaching that observed under congruent conditions ([42]; [63]). In the context of classical color-word Stroop tasks, neural activation patterns exhibit systematic modulation as individuals engage in sustained conflict processing. Specifically, the activation intensity of the ACC in response to conflict tends to decrease across consecutive conflict trials, whereas the involvement of frontal lobe regions increases, reflecting a functional redistribution of cognitive control resources ([25]). This dynamic reconfiguration of neural activity indicates that the individual’s cognitive processing has adapted over time, a phenomenon commonly referred to as the conflict adaptation effect. Drawing on the activation patterns and predictive role of the ACC, researchers have proposed that conflict adaptation arises from enhanced cognitive control triggered by conflict in a preceding trial, thereby facilitating more efficient processing of subsequent conflict-related information.

Building on this foundation, researchers have further extended the scope of conflict adaptation. In the classical color-word Stroop paradigm, conflict arises from the discrepancy between ink color and word meaning. However, to explore whether more complex color-word Stroop tasks can similarly elicit conflict adaptation, some researchers have modified the experimental design to include response-level conflicts, such as conflicting keypress actions. For instance, [48] ([48]) employed a 2:1 mapping paradigm in which participants were instructed to categorize the ink color of color words. Specifically, participants pressed the “Z” key for red or yellow ink and the “M” key for blue or green ink. In the congruent condition, the word meaning and ink color matched (e.g., the word “red” printed in red ink), whereas in the incongruent condition, the word meaning and ink color were associated with opposing response keys (e.g., the word “red” printed in blue ink). By introducing complex color-word Stroop tasks, [48] ([48]) also observed a robust conflict adaptation effect: conflict trials preceded by a conflict trial elicited significantly faster reaction times compared to those following congruent trials. These findings indicate that, within the Stroop paradigm, both classical color-word Stroop tasks ([42]) and complex color-word Stroop tasks ([48]) can reliably induce conflict adaptation effects.

Furthermore, classical and complex color-word Stroop tasks appear to engage distinct neural circuits, offering a novel perspective on the mechanisms underlying conflict adaptation. For instance, [75] ([75]) integrated functional magnetic resonance imaging (fMRI) with a Stroop paradigm to systematically delineate the distinct neural representations associated with these two conflict types. Their findings indicated that classical color-word Stroop tasks primarily recruit the left DLPFC and parietal cortex; however, when task complexity is increased, activation extends to the ACC and the right prefrontal cortex. Notably, the ACC exhibited significantly heightened activation following high-complexity trials, underscoring its central role in conflict detection and subsequent cognitive control processes. More critically, [75] ([75]) proposed that conflicts of varying complexity correspond to distinct stages within the information processing. Classical color-word Stroop tasks are associated with early stages of stimulus recognition and semantic analysis, while color-word Stroop tasks that incorporate response conflict are primarily engaged during later stages involving response selection and execution. This spatial structure and functional attributes dissociation illustrates the cognitive system’s capacity for rapid detection and flexible regulation of conflict through specialized and differentiated neural mechanisms.

It is worth noting that, regardless of whether classical and complex color-word Stroop tasks are employed, prior research has predominantly examined conflict adaptation effects within the same task context ([49]; [9]). Whether there will be cross-task conflict adaptation between different tasks remains an open question. In a pioneering study, [40] ([40]) explored this issue by integrating temporarily ambiguous sentences (e.g., The basketball player accepted the contract that would have to be negotiated) with a Stroop task to examine potential cross-task cognitive control adjustments. The sentence comprehension task was administered using a self-paced reading paradigm, while the Stroop task involved a classical color-word Stroop task. Their findings indicated that the detection of conflict within ambiguous sentences automatically engaged cognitive control mechanisms, thereby facilitating the processing of subsequent incongruent Stroop trials—an effect interpreted as evidence of cross-task conflict adaptation. However, no conflict adaptation effect was observed in the reverse direction: following conflict Stroop trials, reaction times for processing ambiguous and unambiguous sentences did not differ significantly. Nevertheless, using a similar experimental design, [1] ([1]) failed to replicate the cross-task conflict adaptation effects reported by [40] ([40]).

These mixed findings highlight the importance of considering the dynamic nature of cognitive control. Recent studies have indicated that cognitive control is not a static system but one that adjusts flexibly in response to varying task demands and contextual cues ([34]). In particular, the dual mechanisms of control (DMC) framework proposes that cognitive control operates via both proactive and reactive modes, which may differ in their susceptibility to cross-task transfer depending on timing, task structure, and motivational context ([11]; [32]). These theoretical advances suggest that successful cross-task adaptation may depend on whether control states are sustained across tasks or must be reconfigured, which could explain the inconsistent replication of such effects. More recent research has further refined our understanding of cross-task transfer by emphasizing the role of task similarity or cognitive overlap. For instance, [38] ([38]) demonstrated that transfer of control settings is more likely when tasks share common control demands or structural features, even if the stimuli differ. Moreover, [68] ([68]) provided neurophysiological evidence that cross-task transfer is modulated by the degree of shared neural control architecture, particularly within prefrontal networks. Collectively, these findings indicate that the success of cross-task adaptation may depend on the degree to which prior control states are applicable to novel tasks.

Given that the Stroop task employed by [40] ([40]) involved only the classical color-word Stroop task and did not incorporate response conflict, [22] ([22]) extended this line of research by independently manipulating both the classical color-word Stroop task and response conflict tasks to further examine their interaction with the processing of ambiguous sentences. However, [22] ([22]) also failed to replicate the findings of [40] ([40]). Specifically, no significant differences in reaction times were observed between ambiguous and unambiguous sentences following either type of Stroop conflict, nor was there evidence of a reverse transfer effect. Although some researchers have argued that response conflict evokes greater cognitive control demands than the classical color-word Stroop task ([52]; [81]), no conflict adaptation effects were observed. These findings collectively suggest that the existence of cross-task conflict adaptation between Stroop-based cognitive control tasks and sentence comprehension remains contentious and warrants further empirical clarification.

However, some researchers have argued that these studies predominantly rely on behavioral experimental methods and that behavioral measures typically capture only end-point outcomes of task performance rather than providing insights into the underlying cognitive processes ([11]; [36]; [59]). As a result, despite variations across studies, such approaches offer limited explanatory power with respect to the dynamic nature of cognitive adjustment ([11]; [36]; [39]; [77]). In response to these limitations, recent research has increasingly incorporated eye-tracking technology to obtain continuous time-series data ([60]), thereby enabling the investigation of individuals’ attention allocation patterns and processing strategies during conflict-related tasks. For instance, [36] ([36]) employed a cross-task adaptation paradigm that combined a Stroop task with an auditory verbal comprehension task and used eye-tracking techniques to systematically examine the temporal unfolding of conflict adaptation and its influence on language comprehension. Their findings indicated that, following incongruent Stroop trials, participants were able to more rapidly select and fixate on the image corresponding to the sentence meaning in the subsequent auditory comprehension task, thereby reducing fixation errors and enhancing overall processing efficiency.

Extending previous investigations into cross-task cognitive control, [35] ([35]) conducted an eye-tracking study utilizing the Flanker task and demonstrated that the efficiency of cognitive processing during subsequent ambiguous sentence comprehension was significantly enhanced following a conflict trial. Specifically, when participants encountered a high-conflict Flanker trial prior to hearing an ambiguous sentence such as “Put the horse on the binder onto the scarf”, they exhibited shorter response latencies and higher accuracy in selecting the corresponding target image, indicating a heightened capacity for real-time cognitive control during syntactic parsing. Based on the results of these studies, the authors interpreted the findings within the framework of conflict monitoring theory. According to this theory, the cognitive system contains a specialized mechanism that continuously monitors for conflict during information processing. When conflict is detected, particularly in cases where anticipated input diverges from the actual information being processed, the monitoring mechanism becomes engaged and initiates regulatory processes through the allocation of cognitive control resources. The presence of conflict is thus construed as a critical signal indicating increased demands on cognitive control, which prompts the system to mobilize additional resources in order to enhance the efficiency of subsequent cognitive operations ([8]). Within the domain of verbal comprehension, as linguistic input unfolds incrementally, individuals are able to recruit this domain-general cognitive control mechanism to implement real-time adjustments in sentence interpretation, thereby facilitating the resolution of ambiguity and promoting more effective language understanding.

In line with this research trajectory, several scholars have extended the investigation of sentence ambiguity into the domain of thematic role assignment ([69]). Thematic role assignment refers to the cognitive process through which, during sentence comprehension, the predicate verb is systematically mapped onto its corresponding noun phrases, thereby allowing for the attribution of specific semantic roles such as agent and patient and indicating the functions and relationships of the noun phrases in the action or event ([21]). For instance, in interpreting a sentence such as “The boy ate the apple”, readers must efficiently identify “the boy” as the agent performing the action and “the apple” as the patient receiving it. However, in cases of incongruent sentences, such as “The apple ate the boy”, the thematic roles are assigned in a manner that conflicts with real-world knowledge, creating a strong semantic anomaly. This conflict arises not merely from syntactic structure but from the fundamental mismatch between the expected thematic roles and the actual content of the sentence. Critically, the identification of such conflicts in incongruent sentences typically occurs when encountering the verb and its subsequent object, where the expected agent-patient relationship sharply diverges from common world knowledge. For example, in the sentence “The apple ate the boy”, the conflict is immediately triggered upon encountering the verb “ate” and continues through the processing of the object “the boy” because it violates the basic semantic expectation that an apple cannot act as the agent of eating.

Resolving such conflicts requires substantial cognitive control, as the reader must override the automatic expectation based on prior world knowledge to arrive at a coherent interpretation. This aligns with the conflict monitoring theory, which posits that cognitive control mechanisms are recruited to detect and resolve conflicts when the expected cognitive schema is violated ([8]; [23]). In the context of incongruent sentences, the comprehension system is challenged by a mismatch between syntactic structure and real-world semantic expectations. This discrepancy activates conflict monitoring mechanisms that serve to detect and resolve interpretive inconsistencies. Specifically, when the thematic role assignment violates animacy-based plausibility—such as assigning an agent role to an inanimate noun—readers must override the default heuristic that animates are typically agents and inanimates are patients ([26]; [50]). Conflict monitoring theory posits that such violations are registered as cognitive conflicts, which trigger increased cognitive control demands to suppress contextually inappropriate interpretations and update the mental representation accordingly ([8]; [56]). This process is not limited to resolving syntactic ambiguity but extends to thematic processing, wherein the parser actively monitors for semantic implausibility during role assignment. When the initial parse leads to an implausible thematic configuration—as in an inanimate subject performing an action—the system initiates reanalysis mechanisms, reallocating cognitive resources to integrate syntactic and semantic information in a coherent manner ([46]; [73]). In this view, conflict monitoring offers a functional framework for understanding how readers dynamically revise their interpretations in response to implausibility at the level of thematic role assignment. Thus, incongruent sentences elicit conflict signals that engage the broader cognitive control system, particularly in the face of violations that compromise the integration of syntactic form with semantic expectation ([65]).

[69] ([69]) investigated the processing of passive sentences involving thematic role assignment conflicts, such as “The fox was chased by the rabbit”, in which the semantic plausibility favors an interpretation aligned with “The fox chases the rabbit”, whereas the syntactic structure indicates the reverse assignment of roles. The study revealed that when participants were exposed to cognitive conflict in a preceding trial, such as incongruent Stroop stimuli, their performance in subsequent thematic role assignment improved. This facilitation was evidenced by faster and more accurate selection of the picture corresponding to the correct interpretation of the sentence. The observed divergence in gaze trajectories during the processing of conflict sentences indicates that the activation of cognitive control mechanisms facilitates the parsing of thematic role assignments and mitigates errors resulting from semantic interference. These results offer additional empirical support for the conflict monitoring theory by highlighting the modulatory role of cognitive control in resolving syntactic-semantic competition.

Although studies employing eye-tracking methodologies have produced relatively consistent findings, they have predominantly been conducted within the domain of auditory language comprehension. Listening comprehension is an innate human capacity that begins to develop in infancy and follows a biologically driven trajectory ([58]). In contrast, reading comprehension is a learned skill that develops more gradually during childhood and requires explicit instruction and sustained practice. Furthermore, a subset of individuals, such as those with developmental dyslexia, experience persistent difficulties in achieving reading fluency and accuracy. Given these developmental and cognitive differences between listening and reading, the question of whether conflict adaptation occurs in reading comprehension remains unresolved. Existing evidence from self-paced reading paradigms has yielded inconclusive results regarding the presence and reliability of such effects ([1]; [22]; [40]).

Consequently, it is essential to employ alternative methodologies such as eye-tracking in order to capture dynamic and fine-grained indicators of cognitive processing. Eye-tracking techniques offer higher temporal resolution, which makes them particularly well-suited for investigating the time course of conflict adaptation in language comprehension. Importantly, eye-tracking may also help address concerns regarding the non-replicability of cross-task conflict adaptation effects observed in self-paced reading paradigms ([22]; [40]). This is because eye-tracking provides continuous, character-level measures of processing—such as fixation durations and regressions—that are sensitive to subtle variations in cognitive control engagement. In contrast to self-paced reading, which yields only one data point per word or region, eye-tracking captures the real-time dynamics of sentence processing, allowing for more reliable detection of conflict-related effects. Thus, eye-tracking offers a methodological advantage by allowing researchers to pinpoint both the temporal onset and spatial locus of adaptation within the sentence processing stream. This approach may facilitate a more precise examination of the underlying mechanisms and temporal dynamics of conflict adaptation during reading comprehension.

In addition to these methodological considerations, it is important to note that most existing research on conflict adaptation in the reading domain has focused exclusively on the classical color-word Stroop task, with relatively little attention given to the impact of task complexity on conflict adaptation. However, it is well established that Stroop tasks of varying complexity are associated with distinct neural activation patterns, suggesting that task complexity may significantly influence the underlying cognitive control processes involved ([75]). Investigating the differential effects of various types of conflict on sentence comprehension can facilitate a more comprehensive understanding of the underlying mechanisms and dynamic processes of conflict adaptation in language processing.

Moreover, cognitive control plays a critical role in both syntactic parsing and semantic integration during reading comprehension, particularly when individuals are confronted with conflicting cues from multiple information sources ([56]). In this context, the Chinese language presents unique processing demands. As a zero-morphology language, Chinese lacks explicit morphological markers and clear word boundaries that are typically present in alphabetic writing systems ([79]). In addition, the ideographic nature of Chinese characters provides dense visual information at the perceptual level, thereby increasing the cognitive load during reading. These linguistic and orthographic characteristics necessitate that Chinese readers engage more sophisticated and flexible cognitive control mechanisms, particularly when processing syntactic structures involving thematic role assignment. For example, certain Chinese verbs may simultaneously function as predicates and as complements within an embedded clause. Accurate role assignment in such cases depends heavily on contextual interpretation, requiring readers to dynamically monitor, revise, and resolve structural ambiguities. This judgment process typically involves both the initial interpretation and subsequent re-evaluation of conflicting information, thereby requiring readers to engage a high level of conflict adaptation in order to adjust their interpretive strategies and respond efficiently to contextual changes.

The present study employed two experiments to investigate the influence of conflict adaptation on thematic role assignment in Chinese sentence comprehension. Experiment 1 utilized a classical color-word Stroop task to assess basic conflict adaptation effects, while Experiment 2 introduced a more complex Stroop task by incorporating response-level conflict through a 2:1 response mapping paradigm. This manipulation added an additional conflict at the execution stage, allowing us to distinguish how increased task complexity, particularly involving response conflict, influences subsequent language comprehension. The current design allowed for a more comprehensive examination of the impact of task complexity on cognitive control processes during sentence comprehension.

To investigate the impact of cognitive conflict on sentence comprehension, the present study employed an eye-tracking reading paradigm in which participants were visually presented with sentences containing either congruent or incongruent thematic role assignments. By comparing eye movement patterns following Stroop trials that varied in conflict complexity, the study aimed to determine whether prior exposure to cognitive conflict facilitates the processing of syntactic structures involving thematic role ambiguity. This methodological approach allowed for a fine-grained, temporally sensitive analysis of conflict adaptation as it unfolds during real-time sentence processing. Notably, in contrast to previous studies that have primarily relied on auditory comprehension paradigms, the current research focused on visually presented language information within a naturalistic reading context, thereby broadening the methodological scope of conflict adaptation research in language comprehension.

Utilizing eye-tracking methodology, the present study simultaneously examines both early and late stages of sentence processing in order to assess the dynamic impact of conflict adaptation on thematic role assignment. Early-stage processing was indexed by First Fixation Duration (FFD) and Gaze Duration (GD), which primarily reflect lexical access and initial syntactic parsing ([61]). Gaze duration is defined as the sum of all first-pass fixations on a given interest area, from the moment the eyes first land on that region until they move away for the first time. These measures are considered sensitive to early decoding and the detection of incongruent linguistic input ([19]; [84]). Later-stage processing was captured through Second Reading Time (SRT) and Total Reading Time (TRT), which reflect late syntactic integration and higher-level semantic processing. These indicators are commonly interpreted as evidence of syntactic reanalysis and meaning integration, as they index the regulation and repair mechanisms involved in reading comprehension ([17]; [37]). In addition, we also pay attention to the Regression Path Reading time (RPR), which reflects the time spent rereading prior regions of text following comprehension difficulty. This measure is widely considered an indicator of syntactic revision or semantic reinterpretation triggered by conflict detection ([78]).

Drawing on the findings of [69] ([69]), the present study hypothesizes that conflict induced by the Stroop task will elicit a cross-task conflict adaptation effect, which in turn modulates the allocation of cognitive control resources during thematic role assignment. Specifically, it is expected that following exposure to conflict, individuals will exhibit enhanced cognitive control, as reflected in fixation durations during the processing of thematic role assignment. In particular, eye movement patterns in the region of interest are anticipated to be influenced by the congruency of the Stroop condition in the preceding trial ([29]; [43]). Moreover, based on the findings of [16] ([16]), conflict adaptation is hypothesized to exert its strongest effects during later stages of cognitive processing, particularly when semantic integration and syntactic decision-making are required ([8]; [23]). Therefore, the effect is expected to manifest in late eye movement indicators, such as TRT and RPR, with reduced processing durations observed when a Stroop conflict trial precedes a thematic role conflict sentence ([33]; [83]). Finally, given that color-word Stroop tasks involving response conflicts have been shown to elicit stronger and more sustained activation of the cognitive control network than classical color-word Stroop tasks ([52]; [81]), it is further hypothesized that the conflict adaptation effect will emerge earlier in the time course following complex color-word Stroop tasks. Accordingly, participants are expected to exhibit shorter fixation durations in early-stage indicators after experiencing a complex conflict trial, suggesting more rapid deployment of control resources to facilitate syntactic parsing and reduce interference from semantic incongruity ([55]).

## 2. Experiment 1

### 2.1. Participant

In accordance with the methodology outlined by [36] ([36]), the sample size required for this experiment was determined using G*Power (version 3.1.9.7). Following the recommendations of [18] ([18]), an effect size (*f*) of 0.25 (medium), a significance level of 0.05, and a statistical power of 0.80 were applied. The repeated measures analysis of variance suggested that a minimum of 24 participants would be necessary. To enhance the robustness of the results, a total of 44 participants were recruited. During the data analysis phase, three participants were excluded due to reading accuracy rates below 80%, while five additional participants were excluded due to excessive data loss (>25%), attributed to factors such as blinking, body movements, and incorrect responses. As a result, the final sample consisted of 36 participants, with a mean age of 19.97 years (age range: 18–26 years, *SD* = 1.74). All participants were college students, native Chinese speakers, with no history of psychiatric or neurological disorders, normal or corrected-to-normal vision, and right-handedness. Prior to the commencement of the experiment, all participants provided informed consent and were compensated upon completion. The study was conducted according to the requirements of the Declaration of Helsinki ([82]) and ethical approval was obtained from the Human Subjects Protection Committee of Northeast Normal University (approval number: 2024027).

### 2.2. Experimental Design and Materials

The current study employed a 2 (Stroop type: congruent vs. incongruent) × 2 (Sentence type: congruent vs. incongruent) within-subjects experimental design. The experiment comprised four conditions: Stroop congruent-Sentence congruent (C-C); Stroop congruent-Sentence incongruent (C-I); Stroop incongruent-Sentence congruent (I-C); and Stroop incongruent-Sentence incongruent (I-I).

Participants are required to judge the colors of Chinese characters presented on the screen, with the target colors being red, yellow, and green. The experiment consists of 20 congruent trials, in which the meaning of the Chinese characters corresponds with the ink color (e.g., the Chinese character “红 (red)” written in red ink, “黄 (yellow)” in yellow ink, and “绿 (green)” in green ink). Additionally, 20 incongruent trials are included, where the meaning of the Chinese characters does not align with the ink color (e.g., the character “红 (red)” written in yellow ink, “绿 (green)” in red ink, and “黄 (yellow)” in green ink). This setup ensures that the cognitive control mechanisms engaged are identical to those required for the language task ([52]). To further enhance the internal validity of the Stroop task, 20 neutral trials are incorporated, in which the meaning of the Chinese characters bears no relation to the ink color (e.g., the character “玩 (play)” written in red ink or “国 (country)” in green ink). All neutral trials are combined with filler sentences.

For the reading materials related to thematic role assignment, 40 sets of core phrases with a subject-verb-object structure were compiled. An example of such a phrase is: “司机检查轮胎 (The driver checks the tires)”. Subsequently, thematic role assignment statements were constructed based on these core phrases. For instance, the congruent sentence reads: “长途跋涉之后司机检查轮胎观察磨损情况 (After a long journey the driver checks the tires to observe the wear condition)”, while the incongruent sentence is: “长途跋涉之后轮胎检查司机观察磨损情况 (After a long journey the tires checks the driver to observe the wear condition)”. In the congruent sentence, “司机 (Driver)” serves as subject and “轮胎 (Tires)” as object, while their positions are reversed in the incongruent sentence. In the sentence comprehension task, our analysis focused on two-character words occupying the subject position (NP1), verb position (V), and object position (NP2), which were designated as the area of interest. For congruent sentences, the mean stroke numbers for the two-character words in the NP1, V, and NP2 positions were 15.28 (*SD* = 3.40), 15.28 (*SD* = 3.59), and 15.08 (*SD* = 3.59), respectively. The corresponding word frequency values were 5.19 (*SD* = 5.31), 5.07 (*SD* = 5.60), and 5.07 (*SD* = 5.60), with frequency values logarithmically transformed using base 10. Word frequency data were obtained from the Modern Chinese Corpus Database (http://lingua.mtsu.edu/chinese-computing/ (accessed on 10 August 2024)).

To assess the degree of semantic acceptability across sentences, we recruited 20 participants to rate the plausibility of sentences on a 5-point Likert scale, where a score of 5 indicated the highest level of plausibility. Statistical analysis revealed a significant difference in acceptability ratings between congruent and incongruent sentences (*p* < 0.05). The acceptability scores (mean ± *SD*) were 4.57 ± 0.35 and 1.63 ± 0.28, respectively. To reduce the likelihood that participants would form strong expectations regarding the two-character words in the interest area, a preliminary cloze test was administered to 20 students who had not participated in the formal experiment. The purpose of this test was to assess the predictability of core phrases within sentence structures. In the cloze test, participants were presented with the content preceding the two-character word of interest area, such as “长途跋涉之后司机检查”, and were instructed to predict the subsequent two-character word. The successful prediction rates for congruent sentences were 6.6%, indicating that the context of sentences containing the core phrase does not create strong expectations for the critical two characters.

### 2.3. Apparatus

The EyeLink 1000 Plus tracker (SR Research Ltd., Ottawa, ON, Canada) was used to present the materials and record eye movement data with a sampling rate of 1000 Hz. Although reading was binocular, eye movements were recorded only from the right eye, using chin rests to minimize head movements. Sentences were presented on the 26-inch Dell monitor with a frame rate of 144 Hz. The screen resolution was 1024 × 768 pixels and interfaced with a PC at a viewing distance of 80 cm, which ensured the visual angle of a single character was about 1°. Sentences were presented in white, regular script, size 28 font on a black background, and the Stroop items were similarly presented in size 28 font on a black background. The eye movement trajectory of participants was recorded on the computer using a high-speed camera on the EyeLink tracker. Material presentation and eye movement recording were conducted using dedicated software.

### 2.4. Procedure

All participants were measured individually in a soundproof room. Calibration, validation, and drift correction were conducted before the experiment to ensure the accuracy of the eye movement track record. During calibration, nine standard points (white dots) were randomly displayed on the screen. Participants were instructed to focus on each point sequentially until it disappeared (maximum errors < 0.5°). Following calibration, validation was performed, during which the nine standard points were presented again on the screen (maximum errors < 0.5°). The procedure and scale markers were consistent with those used during calibration. If both calibration and validation were successful, drift correction was then carried out. In this step, a reference point appeared on the screen, and participants were asked to focus on it. Once the drift correction was successfully completed, the experiment proceeded. Prior to the formal experiment, participants completed a practice session. In the Stroop task, participants were required to respond to the ink colors, with red, yellow, and green inks corresponding to the keys “S”, “D”, and “F”, respectively. For the sentence processing task, participants are instructed to read and comprehend the sentences silently. Additionally, a total of 20 questions were randomly distributed across the experiment and were based on the specific sentences currently being presented, and participants were asked to judge the correctness of the answers by pressing the corresponding keys on the keyboard. The keys “J” and “K” were assigned to the responses “correct” and “wrong”, respectively. The accuracy of the participants’ responses was used as an indicator of their attentiveness during the task. The entire experiment lasted approximately 40 min.

To rigorously assess whether cognitive control facilitates readers’ comprehension and processing of thematic role assignment sentences, the current study employs a trial-by-trial design and interleaves the Stroop task with the thematic role assignment language comprehension task in a pseudorandom order. During the experiment, an initial fixation point was initially presented at the center of the screen. Once participants fixated on the center, the fixation point was replaced by a Stroop item (Trial n). Participants were required to identify the color of the Stroop item, which remained visible for 1500 milliseconds or until a response was recorded. Subsequently, participants were presented with a thematic role assignment sentence (Trial n + 1). After reading the sentence, participants pressed the space bar to proceed to a blank screen. The duration of the blank screen was randomly set to a value between 1000 and 2000 milliseconds. Once the blank screen ended, the next cycle began (as illustrated in Figure 1). For the sake of clarity, in this study, the combination of Trial n and Trial n + 1 is referred to as a cycle.

To ensure an equal number of trials for each of the four Stroop-sentence combinations (C-C, C-I, I-C, I-I), all materials were arranged using a Latin square design, resulting in two pseudorandomized versions of the experiment. Each version contained a fixed trial sequence that was identical across participants assigned to the same list. This pseudorandomization approach was adapted from prior work (e.g., [36]; [35]; [69]). Since the Latin square design already counterbalanced order effects across conditions, cycle order was not included as a factor in the statistical model. Each version comprised 8 practice cycles and 40 experimental cycles. Additionally, 40 filler cycles were included. The sentence structure of the filler was consistent with that of the experimental sentences, and 20 questions were randomly assigned to both the experimental and filler sentences. Participants completed one of the two versions, with the practice and filler cycles repeated accordingly.

### 2.5. Results

After excluding participants with accuracy rates below the threshold, the reading comprehension accuracy for all remaining participants exceeded 80%. In the Stroop task, the Stroop effect was first measured, revealing a significant classical Stroop effect (congruent condition: *M* = 994 ms, *SD* = 347; incongruent condition: *M* = 1121 ms, *SD* = 389; *p* < 0.001). Data with a single fixation duration either less than 80 ms or greater than 1200 ms ([20]), as well as data points that exceeded three standard deviations from the mean (NP1: 1.58%; V: 1.84%; NP2: 1.72%), were excluded from the analysis. Additionally, if a participant’s response on Trial n of the Stroop task was incorrect, the sentence processing data from the subsequent Trial n + 1 were not included in the analysis, resulting in the exclusion of 20 trials (2.07%). This study focused on five eye movement indicators: FFD and GD reflect early processing stages, and SRT, TRT, and RPR reflect later processing stages. Descriptive statistical results are presented in Table 1.

For each index, a Linear Mixed-Effects Model (LMM; [3]) was employed, utilizing the R programming environment (Version 3.6.3; [62]). The *lmerTest* package (Version 3.1-2; [47]) was used for *p*-values. Continuous dependent variables, including FFD, GD, SRT, TRT, and RPR, were log-transformed ([44]). Data analysis was conducted using the *lme4* package ([5]), with Stroop condition (congruent vs. incongruent) and sentence type (congruent vs. incongruent) as fixed effects and subjects and items as random effects.

During data analysis, categorical variables (such as Stroop type and sentence type) were initially encoded virtually. For the random effects, we incorporated variance components related to both subjects and items for intercepts, as well as random slopes for fixed effects, starting with a full random effects structure. The full model was constructed based on the principle of maximal random effects (including main effects and interactions of Stroop type and sentence type, as well as all random effects for subjects and items; [4]). If the full model failed to converge, random correlation terms were removed. If convergence issues persisted, principal component analysis was employed to assess the variance of random effects within the model. Random effects with a contribution rate equal to or less than zero were deleted, and the model was simplified iteratively until convergence was achieved and no singular fit warnings were issued ([13]). Finally, likelihood-ratio tests were conducted between the convergent model and the null model (excluding fixed factors), and the fixed effects, main effects, and simple effects of the model were calculated.

#### 2.5.1. NP1 Area

The results indicated that the optimal fitting models for FFD, GD, SRT, TRT, and RPR all included fixed effects (Stroop type and sentence type) as well as random effects associated with items and subjects. The final model is presented in Appendix A. The likelihood-ratio tests revealed significant differences between the fitted models and the null models for all indicators except FFD (GD: χ^2^ = 11.881, *df* = 3, *p* < 0.01; SRT: χ^2^ = 12.536, *df* = 3, *p* < 0.01; TRT: χ^2^ = 23.062, *df* = 3, *p* < 0.001; RPR: χ^2^ = 13.014, *df* = 3, *p* < 0.01). These findings suggest that the fitted models provide a superior fit to the data compared to the null models. A summary of the fixed effects in the best-fitting models for these five eye movement indicators is presented in Table 2.

First Fixation Duration: Statistical analyses revealed that neither the main effect of sentence type nor the main effect of Stroop type reached statistical significance. Furthermore, the interaction between sentence type and Stroop type was also nonsignificant.

Gaze Duration: The analysis revealed that only the main effect of sentence type was significant, with reading time for congruent sentences being significantly shorter than incongruent sentences (*b* = −0.049, *SE* = 0.014, *t* = −3.550, *p* < 0.01).

Second Reading Time: The main effect of sentence type and the interaction between Stroop type and sentence type were significant. Simple effects analysis revealed that for congruent sentences, reading time under the IC condition was significantly longer than under the CC condition (*b* = −0.275, *SE* = 0.123, *t* = −2.237, *p* = 0.029). However, for incongruent sentences, no conflict adaptation effect was observed.

Total Reading Time: The main effects of Stroop type and sentence type were significant. The interaction between Stroop type and sentence type approached significance. Simple effects analysis revealed that for congruent sentences, reading time in the IC condition was significantly longer than in the CC condition (*b* = −0.077, *SE* = 0.029, *t* = −2.695, *p* < 0.01). However, for incongruent sentences, no conflict adaptation effect was observed in total reading time.

Regression Path Reading time: The main effects of Stroop type and sentence type were significant. Reading time for congruent sentences was significantly shorter than that for incongruent sentences (*b* = −0.041, *SE* = 0.015, *t* = −2.791, *p* < 0.01). The interaction between Stroop type and sentence type was not significant (*p* > 0.01).

#### 2.5.2. V Area

The results indicated that the optimal fitting models for FFD, GD, SRT, TRT, and RPR all included fixed effects (Stroop type and sentence type) as well as random effects associated with items and subjects. The final model is presented in Appendix A. The likelihood-ratio tests revealed that the best-fitting models for FFD and GD did not significantly differ from the null model (*p*s > 0.5). However, the best-fitting models for the later indicators (SRT: χ^2^ = 28.533, *df* = 3, *p* < 0.001; TRT: χ^2^ = 17.48, *df* = 3, *p* < 0.001; RPR: χ^2^ = 30.884, *df* = 3, *p* < 0.001) were significantly different from the null model. These findings suggest that the fitted models provide a superior fit to the data compared to the null models. A summary of the fixed effects in the best-fitting models for these five eye movement indicators is presented in Table 2.

First Fixation Duration and Gaze Duration: Statistical analyses revealed that neither the main effect of sentence type nor the main effect of Stroop type reached statistical significance. Furthermore, the interaction between sentence type and Stroop type was also nonsignificant. These findings indicate that the conflict adaptation effect did not manifest in the measures associated with early-stage response processing.

Second Reading Time: The main effect of sentence type and the interaction between Stroop type and sentence type were significant. Simple effects analysis revealed that for congruent sentences, reading time under the IC condition was significantly longer than under the CC condition (*b* = −0.365, *SE* = 0.102, *t* = −3.591, *p* < 0.001).

Total Reading Time: The main effect of sentence type was significant, and the interaction between Stroop type and sentence type was also significant. Simple effects analysis revealed that for congruent sentences, reading time in the IC condition was significantly longer than in the CC condition (*b* = −0.057, *SE* = 0.023, *t* = −2.476, *p* = 0.016). In contrast, for incongruent sentences, reading time in the II condition was significantly shorter than in the CI condition (*b* = 0.054, *SE* = 0.023, *t* = 2.352, *p* = 0.022), indicating a conflict adaptation effect (as illustrated in Figure 2).

Regression Path Reading time: The main effect of sentence type was significant, and the interaction between Stroop type and sentence type was also significant. Simple effects analysis revealed that for congruent sentences, reading time in the IC condition was significantly longer than in the CC condition (*b* = −0.076, *SE* = 0.027, *t* = −2.814, *p* = 0.006). In contrast, for incongruent sentences, reading time in the II condition was significantly shorter than in the CI condition (*b* = 0.079, *SE* = 0.027, *t* = 2.949, *p* = 0.004), indicating a conflict adaptation effect.

#### 2.5.3. NP2 Area

The results indicated that the optimal fitting models for FFD, GD, SRT, TRT, and RPR all included fixed effects (Stroop type and sentence type) as well as random effects associated with items and subjects. The final model is presented in Appendix A. The likelihood-ratio tests revealed that the best-fitting models for FFD and GD did not significantly differ from the null model (*p*s > 0.2). However, the best-fitting models for the later indicators (SRT: χ^2^ = 36.20, *df* = 3, *p* < 0.001; TRT: χ^2^ = 27.99, *df* = 3, *p* < 0.001; RPR: χ^2^ = 35.72, *df* = 3, *p* < 0.001) were significantly different from the null model, suggesting that these models provided a superior fit to the data compared to the null model. The fixed effects of the optimal fitting models for the five eye movement indicators are summarized in Table 2.

First Fixation Duration and Gaze Duration: Statistical analyses revealed that neither the main effect of sentence type nor the main effect of Stroop type reached statistical significance. Furthermore, the interaction between sentence type and Stroop type was also nonsignificant. These findings indicate that the conflict adaptation effect did not manifest in the measures associated with early-stage response processing.

Second Reading Time: The main effect of sentence type was significant, and the interaction between Stroop type and sentence type was also significant. Simple effects analysis revealed that for congruent sentences, reading time in the IC condition was significantly longer than in the CC condition (*b* = −0.302, *SE* = 0.097, *t* = −3.104, *p* < 0.01). In contrast, for incongruent sentences, reading time in the II condition was significantly shorter than in the CI condition (*b* = 0.400, *SE* = 0.098, *t* = 4.110, *p* < 0.001), indicating a conflict adaptation effect in second reading time (as illustrated in Figure 3).

Total Reading Time: The main effect of sentence type was significant, and the interaction between Stroop type and sentence type was also significant. Simple effects analysis revealed that for congruent sentences, reading time in the IC condition was significantly longer than in the CC condition (*b* = −0.052, *SE* = 0.022, *t* = −2.326, *p* < 0.05). In contrast, for incongruent sentences, reading time in the II condition was significantly shorter than in the CI condition (*b* = 0.075, *SE* = 0.022, *t* = 3.360, *p* < 0.01), indicating a conflict adaptation effect.

Regression Path Reading time: The main effect of sentence type was significant, and the interaction between Stroop type and sentence type was also significant. Simple effects analysis revealed that for congruent sentences, reading time in the IC condition was significantly longer than in the CC condition (*b* = −0.070, *SE* = 0.029, *t* = −2.458, *p* = 0.017). In contrast, for incongruent sentences, reading time in the II condition was significantly shorter than in the CI condition (*b* = 0.106, *SE* = 0.029, *t* = 3.725, *p* < 0.001), indicating a conflict adaptation effect.

The results of Experiment 1 suggest that thematic role assignment in sentence processing is modulated by cognitive control mechanisms. In both the V and NP2 regions, significant conflict adaptation effects were observed in late-stage indicators, suggesting that readers become aware of the conflict as soon as they encounter the verb. At this point, cognitive control mechanisms are likely activated to suppress inappropriate, prepotent interpretations and facilitate the reanalysis required for accurate thematic role assignment. Interestingly, in the NP1 region, we observed a significant main effect of sentence type on the early eye-tracking indicator of GD. This effect may be attributable to the parafoveal-on-foveal effect, whereby visual or syntactic properties of words located in the parafoveal region—approximately 2 degrees of visual angle from the point of fixation—influence the ongoing processing of the currently fixated (foveal) word ([2]; [41]; [45]). In the present study, when participants encountered an incongruent sentence (e.g., “轮胎检查司机”), they were likely fixating on the subject noun (“轮胎”) while simultaneously previewing the upcoming verb (“检查”) in the parafovea. The thematic incongruity between the inanimate subject and the action-oriented verb may have been detected at this early stage, resulting in longer GD for incongruent sentences compared to congruent ones.

It should be noted that current research has found that after encountering incongruent Stroop trials, reading times for subsequent congruent sentences are significantly prolonged, reflecting a sustained sensitivity to previously encountered conflicts. Given the trial-by-trial design of the current study, this observed effect is also consistent with the principles of conflict adaptation, even in the absence of explicit conflict in congruent sentences. During the processing of conflict trials in the Stroop task, individuals have already engaged cognitive control resources to resolve the conflict, effectively “adapting” to conflict resolution strategies. However, when subsequently encountering a congruent sentence, the cognitive system must readjust its processing strategy to manage the thematic role assignment, a process that may require additional cognitive effort. This need for resource reallocation could account for the longer processing times observed for congruent sentences following conflict trials.

Importantly, the observed reduction in Regression Path Reading time (RPR) for incongruent sentences following incongruent Stroop trials also provides evidence for the facilitation of thematic role assignment. This effect likely reflects a more efficient integration of agent-patient relationships, as cognitive control mechanisms enable the rapid inhibition of prepotent but contextually inappropriate interpretations and the reallocation of attentional resources toward the correct thematic roles. In the context of incongruent sentences, where the expected agent-patient relationship is violated, the ability to quickly suppress conflicting semantic expectations is critical for accurate role assignment. In both the V and NP2 regions, the reduced RPR suggests that readers more effectively resolved the conflict between the thematic expectations and the actual sentence structure, leading to fewer regressions and a smoother integration of sentence elements. This finding aligns with the theoretical perspective that conflict adaptation facilitates the processing of complex linguistic structures by enhancing cognitive control, thereby reducing processing costs during thematic role assignment ([69]; [25]). In Experiment 2, we further investigated whether increasing the complexity of conflict tasks would enhance participants’ sensitivity to conflict adaptation within a cross-task experimental framework.

## 3. Experiment 2

### 3.1. Participant

Similar to Experiment 1, the required sample size was estimated using G*Power (version 3.1.9.7), which determined that a minimum of 24 participants was necessary. To enhance the robustness of the findings, 40 participants were recruited. During data analysis, three participants were excluded due to reading accuracy rates below 80%, while an additional four participants were excluded due to excessive data loss (>25%) resulting from factors such as blinking, body movements, or incorrect responses. Consequently, the final sample comprised 33 participants, with a mean age of 20.46 years (age range: 18–25 years, *SD* = 1.56). All participants were college students and native Chinese speakers with no history of psychiatric or neurological disorders, normal or corrected-to-normal vision, and right-handedness. Prior to the experiment, all participants provided informed consent and received compensation upon completion.

### 3.2. Experimental Design and Materials

The experimental design and thematic role assignment sentences were identical to those in Experiment 1, with the only modification being the introduction of behavioral-level response conflicts to the classical color-word Stroop paradigm, thereby increasing the overall task complexity. Following the methodology of [14] ([14]), the materials for the Stroop task were selected. Specifically, a modified 2:1 mapping paradigm was employed, requiring participants to associate two colors with the same response key. Specifically, the ink colors of the Chinese characters included “red”, “blue”, “yellow”, and “green”, while the corresponding Chinese characters were “红 (red)”, “蓝 (blue)”, “黄 (yellow)”, “绿 (green)”. Participants were instructed to respond based on the ink color of the character by pressing the “D” key for red or blue and the “F” key for yellow or green. In the Stroop task of Experiment 2, two conditions were established: the congruent condition, in which the ink color of the character matched its meaning, and the response-incongruent condition, in which both the ink color and the meaning of the character were incongruent, as were the corresponding response keys.

For the Stroop task, 40 congruent trials were selected in which the Chinese characters “红 (red)”, “蓝 (blue)”, “黄 (yellow)”, and “绿 (green)” were presented in red, blue, yellow, and green ink, respectively. Likewise, 40 incongruent trials were selected, in which the character “红 (red)” was presented in yellow ink, “蓝 (blue)” in green ink, “绿 (green)” in red ink, and “黄 (yellow)” in blue ink. Additionally, 20 trials were designated as neutral conditions, in which the meanings of the Chinese characters had no systematic relationship to the ink color (e.g., “玩 (play)” presented in red ink and “国 (country)” in green ink).

### 3.3. Experimental Apparatus and Procedures

The experimental apparatus and procedures were identical to those in Experiment 1, with the sole exception of the Stroop task.

### 3.4. Result

After excluding participants whose accuracy rates fell below a predefined threshold, the reading comprehension accuracy of all remaining participants exceeded 80%. In the Stroop task, the Stroop effect was first measured, revealing a classic Stroop effect (congruent condition: *M* = 1134 ms, *SD* = 425; incongruent condition: *M* = 1197 ms, *SD* = 426, *p* < 0.01). For the reading comprehension task, we defined the object regions (NP1), verb regions (V), and object regions (NP2) as areas of interest. Fixation durations less than 80 ms or greater than 1200 ms ([20]), as well as data points exceeding three standard deviations from the mean (NP1: 1.58%; V: 1.93%; NP2: 1.72%), were excluded from the analysis. Additionally, if the response to the Stroop task in Trial n was incorrect, the sentence processing data for the subsequent Trial n + 1 were not included in the analysis, leading to the exclusion of 33 trials (2.5% of the total). Five eye movement indicators, including FFD, GD, SRT, TRT, and RPR, were also examined. The descriptive statistics results are presented in Table 3. The data analysis procedure followed the methodology implemented in Experiment 1. Specifically, a linear mixed-effects model was fitted using the R programming environment, and the analysis encompassed the calculation of fixed effects, main effects, and simple effects of the model.

#### 3.4.1. NP1 Area

The results indicated that the optimal fitting models for FFD, GD, SRT, TRT, and RPR all included fixed effects (Stroop type and sentence type) as well as random effects associated with items and subjects. The final model is presented in Appendix B. The likelihood-ratio tests indicated that, with the exception of FFD and GD, the fitted models differed significantly from the null models (SRT: χ^2^ = 37.755, *df* = 3, *p* < 0.001; TRT: χ^2^ = 30.724, *df* = 3, *p* < 0.001; RPR: χ^2^ = 12.969, *df* = 3, *p* < 0.01). These findings suggest that the fitted models provide a superior fit to the data compared to the null models. A summary of the fixed effects in the best-fitting models for these five eye movement indicators is presented in Table 4.

First Fixation Duration and Gaze Duration: Statistical analyses revealed that neither the main effect of sentence type nor the main effect of Stroop type reached statistical significance. Furthermore, the interaction between sentence type and Stroop type was also nonsignificant. These findings indicate that the conflict adaptation effect did not manifest in the measures associated with early-stage response processing.

Second Reading Time: The main effects of Stroop type and sentence type were significant. Notably, reading times for congruent sentences were significantly shorter than those for incongruent sentences (*b* = −0.471, *SE* = 0.067, *t* = −6.988, *p* < 0.001).

Total Reading Time and Regression Path Reading time: Only the main effect of sentence type of TRT and RPR was found to be significant. Reading times for congruent sentences were significantly shorter than those for incongruent sentences (TRT: *b* = −0.109, *SE* = 0.017, *t* = −6.511, *p* < 0.001, RPR: *b* = −0.061, *SE* = 0.016, *t* = −3.829, *p* < 0.001).

#### 3.4.2. V Area

The results indicated that the optimal fitting models for FFD, GD, SRT, TRT, and RPR all included fixed effects (Stroop type and sentence type) as well as random effects associated with items and subjects. The final model is presented in Appendix B. The likelihood-ratio tests revealed that the best-fitting models for FFD and GD did not significantly differ from the null model (*p*s > 0.5). However, the best-fitting models for the later indicators (SRT: χ^2^ = 17.292, *df* = 3, *p* < 0.001; TRT: χ^2^ = 17.626, *df* = 3, *p* < 0.001; RPR: χ^2^ = 30.946, *df* = 3, *p* < 0.001) were significantly different from the null model. These findings suggest that the fitted models provide a superior fit to the data compared to the null models. A summary of the fixed effects in the best-fitting models for these five eye movement indicators is presented in Table 4.

First Fixation Duration and Gaze Duration: Statistical analyses revealed that neither the main effect of sentence type nor the main effect of Stroop type reached statistical significance. Furthermore, the interaction between sentence type and Stroop type was also nonsignificant. These findings indicate that the conflict adaptation effect did not manifest in the measures associated with early-stage response processing.

Second Reading Time: The main effect of sentence type was significant. The interaction between Stroop type and sentence type approached significance. Simple effect analysis did not reveal any statistically significant outcomes.

Total Reading Time: The main effect of sentence type was significant, and the interaction between Stroop type and sentence type was also significant. Simple effects analysis revealed that for congruent sentences, reading time in the IC condition was significantly longer than in the CC condition (*b* = −0.058, *SE* = 0.024, *t* = −2.426, *p* = 0.018).

Regression Path Reading time: The main effect of sentence type was significant, and the interaction between Stroop type and sentence type was also significant. Simple effects analysis revealed that for congruent sentences, reading time in the IC condition was significantly longer than in the CC condition (*b* = −0.075, *SE* = 0.028, *t* = −2.653, *p* = 0.01).

#### 3.4.3. NP2 Area

The results indicated that the optimal fitting models for FFD, GD, SRT, TRT, and RPR all included fixed effects (Stroop type and sentence type) as well as random effects associated with items and subjects. The final model is presented in Appendix B. Likelihood-ratio tests indicated that, with the exception of FFD, where the difference between the best-fitting model and the null model was not statistically significant (*p* > 0.2), all other eye movement indicators exhibited statistically significant differences (GD: χ^2^ = 15.03, *df* = 3, *p* < 0.01; SRT: χ^2^ = 15.92, *df* = 3, *p* < 0.01; TRT: χ^2^ = 22.58, *df* = 3, *p* < 0.001; RPR: χ^2^ = 30.40, *df* = 3, *p* < 0.001). These results suggest that the best-fitting models provided a significantly better fit to the data for these four indicators. A summary of the fixed effects in the best-fitting models for all five eye movement indicators is presented in Table 4.

First Fixation Duration: The statistical analysis revealed that neither the main effects of sentence type and Stroop type nor their interaction reached statistical significance.

Gaze Duration: The main effect of Stroop type was significant. Additionally, the interaction between Stroop type and sentence type was significant. Simple effects analysis revealed that for incongruent sentences, reading time in the II condition was significantly shorter than in the CI condition (*b* = 0.061, *SE* = 0.018, *t* = 3.438, *p* < 0.001), indicating a conflict adaptation effect (see Figure 4).

Second Reading Time: The interaction between Stroop type and sentence type was significant; however, follow-up simple effects analyses did not reveal any statistically significant differences.

Total Reading Time: The main effect of sentence type was significant, and the interaction between Stroop type and sentence type was also significant. Simple effects analysis revealed that for incongruent sentences, reading time in the II condition was significantly shorter than in the CI condition (*b* = 0.064, *SE* = 0.024, *t* = 2.657, *p* < 0.01), indicating a conflict adaptation effect.

Regression Path Reading time: The main effect of sentence type was significant, and the interaction between Stroop type and sentence type was also significant. Simple effects analysis revealed that for congruent sentences only, reading time in the IC condition was significantly longer than in the CC condition (*b* = −0.065, *SE* = 0.029, *t* = −2.257, *p* = 0.027).

The results of Experiment 2 demonstrated that cognitive control mechanisms triggered by the complex color-word Stroop tasks also influenced subsequent sentence processing. In the NP2 region, where the demands for cognitive resource allocation are significantly higher due to increased task complexity, the experiment not only replicated the primary finding from Experiment 1—namely, a significant conflict adaptation effect observed in late-stage indicators within the interest area—but also revealed that this effect emerged earlier, as evidenced by its presence in the early-stage indicator of Gaze Duration. However, in the V region, Experiment 2 did not reveal significant conflict adaptation effects for incongruent sentences, which stands in contrast to the findings of Experiment 1. This discrepancy can be understood from the perspective of cognitive load and resource allocation. Complex color-word Stroop tasks, which introduce response-level conflicts in addition to the classical color-word interference, impose a greater cognitive burden on participants, requiring more extensive resource allocation for response selection and motor control. As a result, participants may have already expended significant cognitive resources during the early stages of processing, leaving fewer resources available for rapid conflict resolution at the verb position. In other words, while more complex tasks demand a higher overall level of cognitive control, they may simultaneously limit the availability of resources for localized conflict adaptation in specific regions, such as the verb area. These findings suggest that as task complexity increases, individuals exhibit greater efficiency in cognitive resource allocation, thereby accelerating the processing of conflicting information.

### 3.5. Cross-Experiment Comparisons

To further assess the effectiveness of the cognitive control tasks with varying levels of complexity used in the present study for influencing thematic role assignment, a cross-experiment comparison was conducted. Specifically, a 2 (Experiment: Experiment 1 vs. Experiment 2) × 2 (Stroop Type: congruent vs. incongruent) × 2 (Sentence Type: congruent vs. incongruent) mixed experimental design was employed to examine the combined effects of task complexity, Stroop type, and sentence type on participants’ sentence processing. The relevant results are presented in Table 5. The result suggests that the impact of Stroop-induced cognitive control on sentence processing is modulated by both the complexity of the task. In the NP1 region, this modulation is evident in both early and late eye-tracking indicators, whereas in the NP2 region, it emerges only in the late-stage indicator of Second Reading Time (SRT). These findings underscore the nuanced role of task complexity in shaping cross-task transfer of cognitive control.

## 4. General Discussion

The findings of the present study not only confirmed that conflict elicited by cognitive control can exert cross-task influence on language processing but also, for the first time, revealed the specific manifestation of conflict adaptation in the domain of thematic role assignment in Chinese. Cross-task conflict adaptation effects were observed following color-word Stroop tasks with varying levels of task complexity. Furthermore, the complex color-word Stroop tasks elicited a stronger pattern of cognitive resource recruitment compared to the classical color-word Stroop tasks, suggesting that greater cognitive control was required to regulate subsequent linguistic processing following exposure to response-related conflicts with greater task demands ([23]; [43]). In addition, during Chinese sentence reading, conflict adaptation significantly modulated the parsing of sentences involving incongruent thematic role assignments. When participants experienced conflict in the preceding trial, eye movement data revealed faster syntactic parsing and fewer regressive behaviors, indicating a marked enhancement in language processing efficiency.

### 4.1. The Role of Classical Color-Word Stroop Task in Cross-Task Conflict Adaptation

Experiment 1 employed the classical color-word Stroop task to investigate the role of cognitive control in thematic role assignment. The results revealed a conflict adaptation effect, which was primarily observed in late-stage eye movement indicators within the interest area. Specifically, processing times in the II condition were significantly shorter than those in the CI condition, as reflected by reduced SRT, TRT, and RPR. Previous research has demonstrated that cognitive load or prior conflict engagement can influence subsequent language processing ([56]). For instance, in a syntactic ambiguity resolution task, [36] ([36]) found that participants who completed a Stroop task subsequently made fewer comprehension errors, suggesting that the engagement of cognitive control mechanisms facilitated more efficient syntactic parsing. Similarly, [35] ([35]) employed the Flanker task in combination with the Visual World paradigm in eye-tracking methodology to investigate whether the engagement of cognitive control facilitates the resolution of incongruent information during listening comprehension. The results indicated that when cognitive control was activated prior to the presentation of incongruent sentences, participants made fewer comprehension errors. Consistent with these findings, the present study further confirmed that the processing of thematic role assignments was modulated by the conflict condition in the preceding trial.

However, previous studies employing the Visual World paradigm have primarily focused on the processing of ambiguous sentences, where participants listened to ambiguous sentences while viewing visual scenes containing both target and distractor images. Eye-tracking data revealed that conflict adaptation effects were primarily observed during the processing of ambiguous sentences. In contrast, for unambiguous (or congruent) sentences, no significant adaptation effects were observed ([36]; [35]). One possible reason for this difference is that these studies, which employed the Visual World paradigm, primarily focused on participants’ fixation proportions toward the correct target image. This design emphasizes the initial stages of auditory sentence processing and may not capture the deeper, more sustained processing required for ambiguous sentences. In contrast, the present study examined the processing of congruent and incongruent thematic role assignment sentences at the visual level using an eye movement reading design, which likely engages a more integrated and continuous conflict monitoring response, potentially extending beyond the momentary effects captured in purely auditory paradigms. This difference in design may explain why the current study observed prolonged reading times for congruent sentences following incongruent trials. It is possible that the cognitive system requires additional effort to shift from a heightened conflict monitoring state, induced by the preceding incongruent trial, back to a more routine processing mode, resulting in the observed processing delay for congruent sentences.

To further clarify how conflict monitoring applies to implausible thematic role assignment, it is essential to delineate the nature of the initial parse. During incremental sentence comprehension, readers typically rely on syntactic cues and semantic heuristics—such as animacy and verb bias—to make rapid, probabilistic assignments of thematic roles ([50]). In the current study, when the subject noun phrase (e.g., “轮胎”, tire) is inanimate, yet the associated verb (e.g., “检查”, check) strongly cues an agentive role. This mismatch likely leads the parser to initially assign an agent role to the inanimate subject based on surface syntactic structure while simultaneously triggering a plausibility evaluation based on semantic constraints. When these two sources of information conflict, the system must re-evaluate the initial parse—a process mediated by conflict monitoring mechanisms ([8]; [73]). Under this view, the initial assignment refers to the syntactically driven interpretation in which the first noun is interpreted as the agent due to its sentence-initial position. However, the semantic implausibility of an inanimate agent violates world knowledge, prompting the conflict monitoring system to initiate reanalysis and suppress the default, but contextually inappropriate, interpretation ([54]). The prolonged reading times observed in our eye-tracking data—particularly in SRT and RPR indicators—reflect this cognitive effort to revise and update the mental model.

Moreover, Experiment 1 extended both the scope and temporal trajectory of conflict adaptation within the classical color-word Stroop task, demonstrating that such adaptation can generalize across tasks from the Stroop task to language processing. This finding suggests a broad transferability of cognitive control mechanisms across distinct processing domains. However, the present results diverge from those of prior studies on cross-task conflict adaptation. For example, [1] ([1]) and [22] ([22]) utilized a self-paced reading paradigm to investigate conflict adaptation effects from syntactic to non-syntactic domains and from perceptual to linguistic domains, respectively, but failed to observe significant cross-task effects. Similarly, [64] ([64]), employing the same paradigm, examined conflict adaptation from the Stroop task to sentence reading and also reported null findings. One plausible explanation for these inconsistencies lies in the methodological limitations associated with the self-paced reading paradigm. Specifically, this method relies exclusively on reaction-time-based measures and captures only the initial pass reading duration, without access to more nuanced indices such as regressions or rereading. Consequently, its relatively coarse temporal resolution may limit the capacity to detect subtle, time-sensitive modulations in cognitive control during sentence comprehension ([39]). By contrast, the present study employed eye-tracking technology, which enables a more fine-grained and temporally sensitive examination of the dynamic regulation of cognitive control during the resolution of thematic role conflicts and provides evidence with higher temporal resolution than single behavioral measurements.

[69] ([69]) investigated whether cognitive control elicited by a preceding color-word Stroop task would influence listeners’ eye movement trajectories during thematic role assignment in auditory sentence comprehension, particularly in the process of identifying the correct referent. Notably, their study employed a visual-world paradigm in which each sentence was paired with corresponding pictures. To enhance referential clarity, all nouns in the object position were animate entities, thereby ensuring a high degree of visual discriminability between agents and patients. In contrast, the present study examined the role of cognitive control in thematic role assignment during silent reading, a modality that recruits distinct perceptual and cognitive mechanisms compared to auditory comprehension. The current design was informed by the well-established linguistic principle that animacy serves as a strong cue in role assignment—where animate nouns are typically interpreted as agents and inanimate nouns as patients ([27]). To maximize thematic conflict, inanimate nouns were systematically placed in the object position, and subject-object roles were reversed across congruent and incongruent sentence conditions. This methodological approach allowed for the induction of robust role conflict while minimizing perceptual confounds. Crucially, it ensures that any observed effects on sentence comprehension stem from genuine cognitive control engagement during thematic processing, rather than from surface-level semantic anomalies.

By leveraging the high ecological validity and temporal resolution of eye-tracking methodology, the current study demonstrated a dynamic relationship between the color-word Stroop task and language processing. Importantly, the effects of cognitive regulation during reading were primarily observed in the reprocessing stage, which is closely associated with comprehension monitoring. In the case of incongruent sentences, this process becomes particularly critical, as readers must not only resolve temporary syntactic ambiguities but also reconcile conflicts between semantic expectations and actual sentence content. This requires the engagement of cognitive control to inhibit the prepotent, contextually inappropriate interpretations and to redirect attention toward more plausible alternative structures. The reduced fixation time observed in the current study for incongruent sentences following incongruent Stroop trials suggests that cognitive control resources were successfully deployed to streamline the integration of unexpected thematic roles, thereby reducing processing difficulty and enhancing overall comprehension efficiency.

### 4.2. The Role of Complex Color-Word Stroop Tasks in Cross-Task Conflict Adaptation

To extend the findings from Experiment 1, Experiment 2 further investigated the role of complex color-word Stroop tasks, which incorporated response conflicts at the behavioral level to increase overall task complexity. The results in the NP2 region not only successfully replicated the conflict adaptation effect but also demonstrated that the complex color-word Stroop tasks more strongly activated the cognitive control network ([52]; [80]). As a result, cognitive control was more sensitively activated following these more demanding Stroop tasks, leading to an earlier onset of conflict adaptation effects. Notably, this effect was detectable at an earlier processing stage, such as GD.

This temporal shift from late-stage indicators in Experiment 1 to earlier-stage markers in Experiment 2 suggests that increased task demands may accelerate the deployment of cognitive control mechanisms. Theoretically, an individual’s proactive control becomes more dominant under high-demand contexts. Under such conditions, control mechanisms are activated in advance and sustained throughout the task, facilitating earlier conflict detection and resolution. Conversely, in low-demand situations, cognitive control is recruited reactively and may appear later in processing. Examining how cognitive control flexibly adapts to variations in task demands contributes to a more nuanced understanding of the mechanisms underlying conflict adaptation. For instance, [28] ([28]) explored the task dependence of conflict adaptation and found that the dynamic adjustment of cognitive control is modulated by the processing demands of the primary task. Specifically, within the context of the Simon task, they observed that higher task demands elicited more pronounced conflict adaptation effects, whereas under low-demand conditions, the adaptation was relatively attenuated ([28]). This pattern is consistent with the present findings, where more complex tasks imposed greater demands on cognitive control, thereby resulting in an earlier and more robust manifestation of conflict adaptation, as evidenced by the temporal advancement of its onset.

When complex color-word Stroop tasks are employed, the influence of conflict adaptation appears to extend beyond the late stage of syntactic integration and also modulates earlier stages of preliminary syntactic analysis. Under conditions of increased task complexity, participants exhibited significantly shorter GD when processing sentences involving incongruent thematic role assignments, suggesting that cognitive control resources were recruited more rapidly to facilitate sentence parsing. This finding is consistent with the results reported by [14] ([14]), who demonstrated that adding response conflicts to a classical color-word Stroop task significantly activates the prefrontal cognitive control network. Such activation enables individuals to more swiftly mobilize regulatory resources in response to conflict and to maintain an elevated state of cognitive control over a prolonged period. In a similar vein, studies utilizing the Flanker task have shown that more complex conflict tasks can sustain heightened sensitivity within the cognitive control system, thereby mitigating interference effects in subsequent trials ([74]; [80]).

Interestingly, no significant conflict adaptation effects were observed in the verb region for incongruent sentences in the present experiment. This finding suggests a potential shift in participants’ cognitive resource allocation strategies as task complexity increases. Specifically, under heightened cognitive demands—such as those introduced by the preceding complex Stroop task—participants may adopt a more selective and efficient approach to processing sentence components. When encountering sentences with incongruent thematic role assignments, semantic conflict is more fully realized and resolved only upon processing the post-verbal noun phrase. As a compensatory strategy, participants may have prioritized the deployment of cognitive control mechanisms at the NP2 region, where semantic conflict becomes more explicit and integration demands are highest. This pattern suggests that under conditions of heightened cognitive demand, individuals strategically allocate processing resources to the sentence region where conflict resolution is most essential. Consequently, this adaptive reallocation of resources may account for the absence of observable conflict adaptation effects in the verb region. These findings are consistent with those reported by [66] ([66]), who demonstrated that when cognitive demands are high, as in tasks involving both semantic retrieval and interference resolution, resource reallocation may preferentially support regions of maximal integration demand, rather than uniformly modulating processing across all regions of the sentence.

It is also important to note that the observed differences between Experiments 1 and 2 should be interpreted with caution; it is crucial to consider the role of task complexity in modulating cognitive control dynamics. Although the theoretical framework of conflict monitoring does not explicitly predict variability across different Stroop paradigms, the present findings suggest that differences in task complexity may elicit distinct patterns of control engagement. Moreover, these differences should be regarded as post-hoc interpretations, given that the study was not originally designed to systematically compare the two Stroop paradigms. Nonetheless, the observed pattern highlights the importance of considering task-induced variability when evaluating cross-task generalization of conflict adaptation. Future studies should directly manipulate Stroop task complexity within a unified experimental framework to disentangle whether the observed differences stem from variations in control demands or other extraneous factors such as fatigue, strategic adjustments, or stimulus familiarity.

### 4.3. The Mechanism of Cross-Task Conflict Adaptation in Thematic Role Assignment in Chinese

Differences in the neural mechanisms underlying classical and complex color-word Stroop tasks may lead to distinct effects on subsequent cognitive control processes. Empirical research has demonstrated that classical Stroop tasks primarily involve interference at the level of information representation, such as the competition between word meaning and ink color. This form of conflict is associated with activation in the left DLPFC and parietal regions. In contrast, complex Stroop tasks involve competition between alternative response choices and are predominantly linked to activation in the ACC and the right prefrontal cortex ([75]). Given the central role of the ACC in conflict monitoring and cognitive control, complex Stroop tasks involving response-level conflict engage this region more directly. Such engagement enhances the detection of conflict-related signals and promotes broader recruitment of the prefrontal control network ([14]). While classical Stroop tasks also require cognitive regulation, their influence is largely confined to the level of semantic processing, and the associated control mechanisms tend to be comparatively moderate. Consequently, more complex color-word Stroop tasks elicit a stronger engagement of cognitive control resources, which facilitates the inhibition of incorrect responses and the rapid adjustment of task strategies ([80]). Once a broader prefrontal network is activated, the enhanced control mechanisms can be sustained over time. In the context of the present study, this sustained engagement was reflected in participants’ improved efficiency in resolving subsequent thematic role conflicts during sentence processing.

Although the primary aim of the present study was to examine whether the processing of sentences involving incongruent thematic role assignments would be modulated by the congruency of the preceding Stroop task, an additional and noteworthy pattern was observed for congruent sentences. Specifically, congruent sentences that followed incongruent Stroop trials elicited longer reading times than those preceded by congruent Stroop trials. This effect was evident in late-stage eye movement measures across both classical and complex color-word Stroop tasks. Comparable phenomena have been reported in prior research on language processing. For instance, [70] ([70]) found that encountering syntactic conflict in earlier trials influenced the parsing of subsequent sentences, resulting in more effortful reanalysis. Similarly, [15] ([15]), in their investigation of syntactic ambiguity resolution, demonstrated that participants exhibited significantly increased rates of regression following trials involving syntactic conflict. These findings collectively suggest that prior exposure to conflict can modulate subsequent sentence processing by prompting the deployment of additional cognitive resources for monitoring and repair. In light of these observations, it may be inferred that experiencing conflict in a preceding trial induces a form of conflict-resolution inertia. Consequently, even when the subsequent trial does not involve explicit conflict, readers may still engage in heightened reprocessing and exhibit elevated regression behavior, reflecting a carryover of heightened cognitive engagement from the prior trial.

In the processing of thematic role assignment, morphologically rich alphabetic languages typically rely on overt inflectional markers to indicate argument roles, while word order serves a comparatively secondary function. For instance, [30] ([30]), employing a self-paced reading paradigm, demonstrated that when word order cues in Spanish were misaligned with the verb’s argument structure, readers exhibited significantly prolonged reading times for the second noun phrase. A follow-up study by [31] ([31]), using eye-tracking methodology, further revealed that such incongruence between syntactic and argument structure cues resulted in increased fixation durations and more frequent regressions in the post-verbal region. In contrast to morphologically rich alphabetic languages, Chinese lacks overt morphological markers and relies predominantly on canonical Subject-Verb-Object (SVO) word order to convey syntactic relations. Consequently, resolving conflicts in thematic role assignment in Chinese may impose greater demands on cognitive control resources. By investigating how cross-task conflict adaptation modulates thematic role assignment during Chinese sentence processing, the present study demonstrates that, despite the absence of morphological cues, Chinese readers exhibit processing patterns that resemble those observed in alphabetic languages. These findings not only provide empirical support for the cross-linguistic generalizability of cognitive control mechanisms but also highlight the critical role of conflict adaptation in syntactic parsing within a morphologically sparse language like Chinese.

Against the backdrop of existing literature, the present study is the first to investigate the role of conflict adaptation in thematic role assignment in Chinese, a language characterized by the absence of overt morphological markers. In addition, the study employed color-word Stroop tasks of varying complexity to examine their distinct effects on the processing of visually presented linguistic input. The findings provide empirical support for the conflict monitoring theory proposed by [8] ([8]), which posits that when conflict arises between competing cognitive or response processes during task execution, the brain detects this conflict and initiates a regulatory signal to enhance cognitive control, thereby improving performance on subsequent tasks. Given that the conflict adaptation effect is modulated by task complexity, the present study advances our understanding of how cognitive control mechanisms operate in the context of cross-task transfer and provides novel empirical evidence for how cognitive control dynamically regulates sentence processing.

Beyond their theoretical contributions to models of cognitive control and sentence processing, the present findings also offer important cross-linguistic and applied implications. Although the current study was conducted in Chinese—a language that lacks overt morphological markers and relies heavily on word order and semantic plausibility for syntactic interpretation—the mechanisms underlying conflict adaptation may extend to languages with different grammatical architectures. For example, in morphologically rich languages such as German, Russian, or Turkish, syntactic relations are often marked through inflectional morphology rather than word order. In such systems, the timing and locus of conflict detection and resolution may differ due to the earlier availability of syntactic cues ([7]; [6]). Future studies could examine whether conflict adaptation effects emerge at comparable processing stages across language types or whether language-specific features modulate the efficiency and timing of control engagement. Cross-linguistic investigations may thus provide critical insights into the universality versus specificity of cognitive control mechanisms in sentence comprehension.

Nonetheless, several limitations of the present study warrant consideration. First, all nouns in the object position were inanimate; while this manipulation effectively heightened structural incongruence within the experimental framework, it simultaneously restricted the examination of animacy effects on thematic role assignment. Prior research has demonstrated that animacy exerts a robust influence on sentence processing, with animate referents more readily interpreted as agents due to their higher semantic plausibility within event structures ([27]; [57]). The systematic exclusion of animate nouns from object positions thus limits the generalizability of the current findings and prevents a more refined understanding of how animacy may interact with cognitive control mechanisms during conflict resolution. Future research would benefit from a more comprehensive manipulation of animacy across both subject and object positions, allowing for an exploration of its potential moderating role in conflict adaptation processes. Moreover, recent research on conflict adaptation has increasingly shifted its focus toward the question of domain-general adaptation, specifically examining whether conflict adaptation can transcend task-specific boundaries and generalize across distinct cognitive domains ([24]). In the current study, both the color-word Stroop task and the sentence comprehension task involved linguistic materials, thereby limiting the investigation to domain-specific conflict adaptation. Future research should aim to examine the mechanisms of conflict adaptation within a broader range of cognitive contexts. For instance, employing non-linguistic conflict tasks or extending the investigation to domains involving emotional or affective conflict may yield novel insights into the domain-general transferability of conflict adaptation across tasks. In addition, the current study balanced the experimental materials within each version but did not employ a fully randomized presentation. To enhance internal validity and minimize potential order effects, future research should adopt a fully randomized design in the allocation of stimuli across trials and participants.

## 5. Conclusions

The present study employed color-word Stroop tasks of varying complexity to systematically investigate the influence of cognitive control on thematic role assignment in Chinese. The results demonstrated that cognitive control, as elicited by the Stroop paradigm, can exert cross-task modulatory effects on subsequent language processing. Furthermore, as the intensity of cognitive conflict increased, the conflict adaptation effect emerged earlier, indicating that heightened cognitive control demands promote more rapid and efficient adaptive adjustments. These findings expand the theoretical applicability of the conflict monitoring theory and offer novel empirical evidence for its underlying mechanisms in the context of cross-task conflict adaptation.

## Figures and Tables

**Figure 1 behavsci-15-00899-f001:**
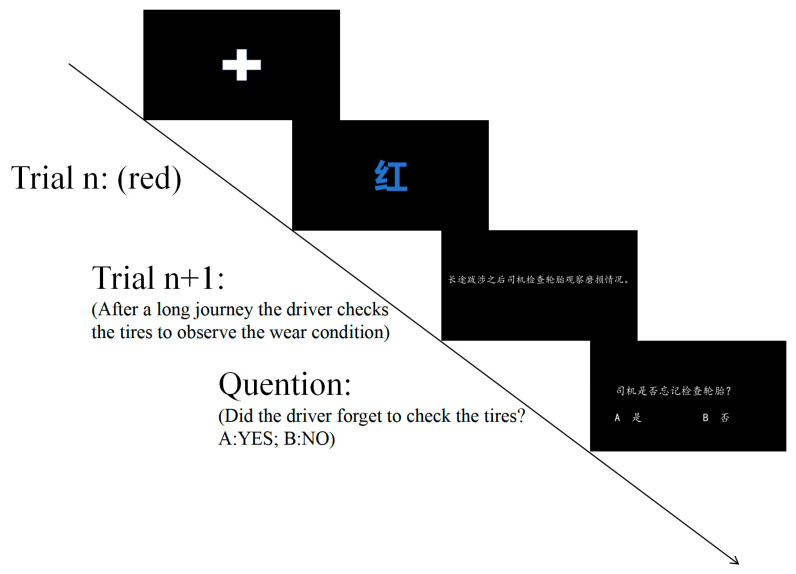
Flowchart of Experiment 1.

**Figure 2 behavsci-15-00899-f002:**
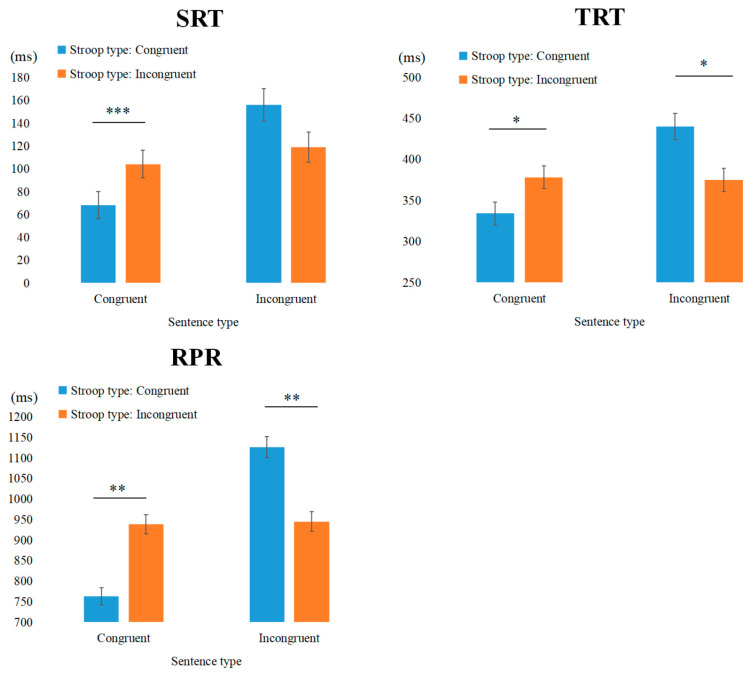
Mean fixation duration (in milliseconds) on the V area in Experiment 1 as a function of preceding Stroop congruency and current sentence type. * indicates *p* ≤ 0.05; ** indicates *p* ≤ 0.01; *** indicates *p* ≤ 0.001; Error bars indicate 95% confidence intervals.

**Figure 3 behavsci-15-00899-f003:**
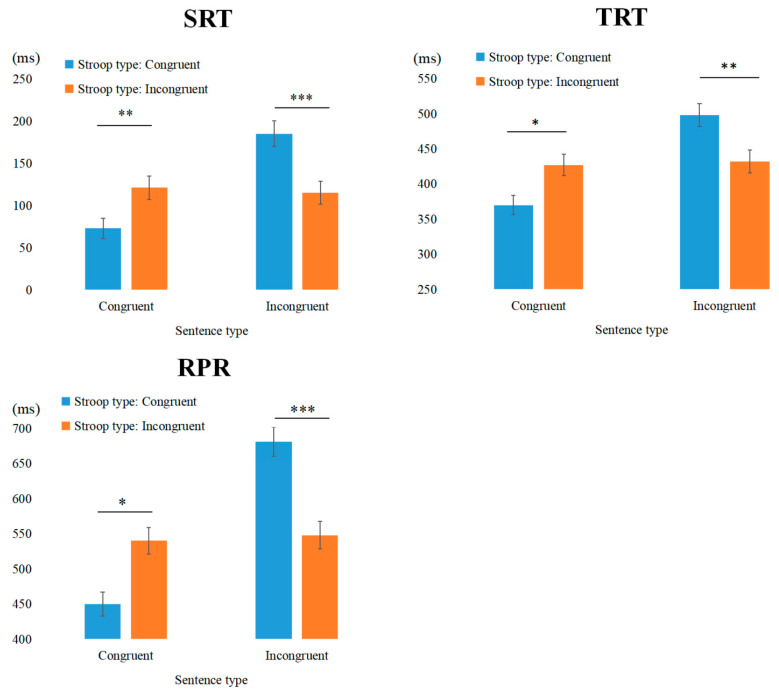
Mean fixation duration (in milliseconds) on the NP2 area in Experiment 1 as a function of preceding Stroop congruency and current sentence type. * indicates *p* ≤ 0.05; ** indicates *p* ≤ 0.01; *** indicates *p* ≤ 0.001; Error bars indicate 95% confidence intervals.

**Figure 4 behavsci-15-00899-f004:**
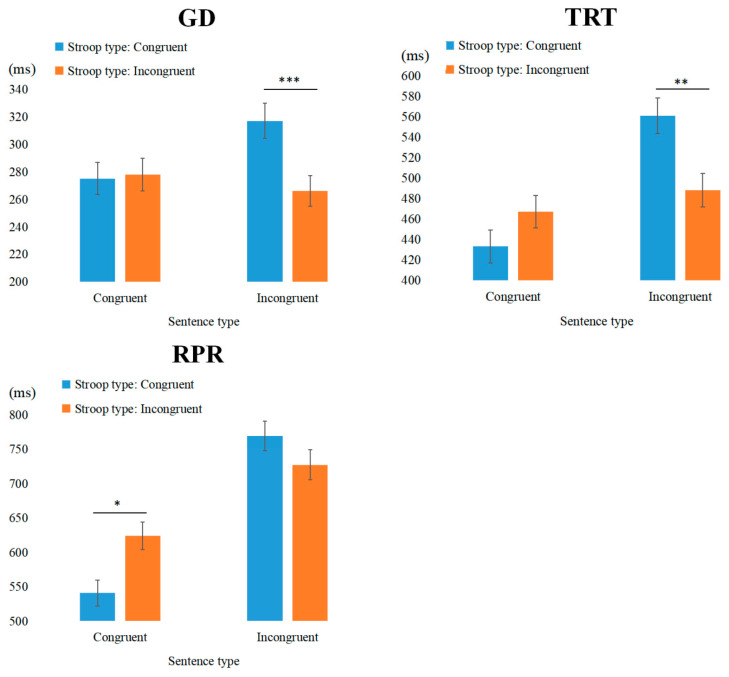
Mean fixation duration (in milliseconds) on the NP2 area in experiment 2 as a function of preceding Stroop congruency and current sentence type. * indicates *p* ≤ 0.05; ** indicates *p* ≤ 0.01; *** indicates *p* ≤ 0.001; Error bars indicate 95% confidence intervals.

**Table 1 behavsci-15-00899-t001:** The mean values (ms) and standard deviations of eye movement indicators within the three interest areas in Experiment 1.

		FFD		GD		SRT		TRT		RPR	
	Condition	*M*	*SD*	*M*	*SD*	*M*	*SD*	*M*	*SD*	*M*	*SD*
NP1	C-C	208	66	278	141	156	223	425	265	358	203
	C-I	218	69	325	171	253	298	563	337	427	234
	I-C	215	67	292	150	220	265	501	278	409	207
	I-I	214	64	321	163	207	257	535	306	420	217
V	C-C	220	66	256	106	68	142	334	185	763	424
	C-I	222	73	266	119	156	205	440	259	1126	664
	I-C	225	69	263	109	104	149	378	192	938	537
	I-I	229	72	256	106	119	176	375	201	945	567
NP2	C-C	219	71	292	130	73	143	370	192	450	300
	C-I	223	71	306	151	185	226	498	262	681	438
	I-C	219	70	298	148	121	187	427	237	540	354
	I-I	222	73	290	137	115	188	432	260	548	381

Note: FFD = First Fixation Duration; GD = Gaze Duration; SRT = Second Reading Time; TRT = Total Reading Time; RPR = Regression Path Reading Time. Means (*M*) and standard deviations (*SD*) were calculated based on the raw data. All mean values are reported in milliseconds (ms).

**Table 2 behavsci-15-00899-t002:** The fixed effects of the linear mixed-effects model for various eye movement indicators in the interest area in Experiment 1.

		NP1					V					NP2				
		*b*	*SE*	*t*	*p*	*CI*	*b*	*SE*	*t*	*p*	*CI*	*b*	*SE*	*t*	*p*	*CI*
FFD	Intercept	2.31	0.008	289.351	<0.001	[2.294, 2.326]	2.329	0.008	293.205	<0.001	[2.313, 2.345]	2.322	0.009	260.273	<0.001	[2.304, 2.340]
	Stroop	0.005	0.008	0.692	0.493	[−0.011, 0.021]	0.011	0.009	1.176	0.250	[−0.007, 0.029]	−0.001	0.007	−0.010	0.992	[−0.015, 0.013]
	Sentence	0.011	0.007	1.444	0.158	[−0.003, 0.025]	0.003	0.008	0.418	0.679	[−0.013, 0.019]	0.006	0.008	0.738	0.465	[−0.010, 0.022]
	Stroop:Sentence	−0.021	0.014	−1.427	0.154	[−0.048, 0.006]	0.005	0.018	0.255	0.801	[−0.030, 0.040]	−0.002	0.017	−0.127	0.900	[−0.035, 0.031]
GD	Intercept	2.423	0.016	149.394	<0.001	[2.392, 2.454]	2.378	0.012	200.829	<0.001	[2.354, 2.402]	2.425	0.016	151.493	<0.001	[2.394, 2.456]
	Stroop	0.009	0.013	0.748	0.461	[−0.016, 0.034]	0.001	0.010	0.018	0.985	[−0.019, 0.021]	−0.010	0.011	−0.905	0.371	[−0.032, 0.012]
	Sentence	0.049	0.014	3.556	**<0.01**	[0.022, 0.076]	0.001	0.010	0.119	0.906	[−0.019, 0.021]	0.004	0.011	0.370	0.714	[−0.018, 0.026]
	Stroop:Sentence	−0.017	0.026	−0.677	0.505	[−0.068, 0.034]	−0.021	0.024	−0.874	0.390	[−0.068, 0.026]	−0.020	0.026	−0.763	0.450	[−0.071, 0.031]
SRT	Intercept	1.342	0.099	13.524	<0.001	[1.148, 1.536]	0.905	0.077	11.712	<0.001	[0.754, 1.056]	0.929	0.088	10.534	<0.001	[0.757, 1.101]
	Stroop	0.058	0.063	0.927	0.354	[−0.065, 0.181]	0.083	0.061	1.357	0.175	[−0.037, 0203]	−0.049	0.067	−0.731	0.469	[−0.180, 0.082]
	Sentence	0.192	0.067	2.861	**<0.01**	[0.061, 0.323]	0.306	0.069	4.444	**<0.001**	[0.171, 0.441]	0.307	0.069	4.472	**<0.001**	[0.172, 0.442]
	Stroop:Sentence	−0.433	0.211	−2.048	**0.048**	[−0.847, −0.019]	−0.565	0.163	−3.476	**0.001**	[−0.884, −0.246]	−0.703	0.141	−4.993	**<0.001**	[−0.979, −0.427]
TRT	Intercept	2.621	0.025	103.289	<0.001	[2.572, 2.670]	2.515	0.017	147.812	<0.001	[2.482, 2.548]	2.565	0.021	119.851	<0.001	[2.524, 2.606]
	Stroop	0.029	0.015	1.972	**0.056**	[0.000, 0.058]	0.001	0.012	0.119	0.906	[−0.023, 0.025]	−0.012	0.015	−0.757	0.454	[−0.041, 0.017]
	Sentence	0.074	0.017	4.470	**<0.001**	[0.041, 0.107]	0.050	0.015	3.307	**0.003**	[0.021, 0.079]	0.063	0.015	4.100	**<0.001**	[0.034, 0.092]
	Stroop:Sentence	−0.096	0.049	−1.955	**0.058**	[−0.192, 0.000]	−0.110	0.038	−2.879	**0.007**	[−0.184, −0.036]	−0.127	0.033	−3.885	**<0.001**	[−0.192, −0.062]
RPR	Intercept	2.535	0.020	121.478	<0.001	[2.496, 2.574]	2.885	0.027	106.844	<0.001	[2.832, 2.938]	2.645	0.027	97.644	<0.001	[2.592, 2.698]
	Stroop	0.031	0.014	2.235	**0.032**	[0.004, 0.058]	−0.002	0.016	−0.122	0.911	[−0.033, 0.029]	−0.018	0.016	−1.099	0.279	[−0.049, 0.013]
	Sentence	0.039	0.014	2.725	**0.012**	[0.012, 0.066]	0.082	0.016	4.966	**<0.001**	[0.051, 0.113]	0.088	0.016	5.366	**<0.001**	[0.057, 0.119]
	Stroop:Sentence	−0.058	0.041	−1.413	0.166	[−0.138, 0.022]	−0.155	0.044	−3.533	**0.001**	[−0.241, −0.069]	−0.176	0.047	−3.784	**<0.001**	[−0.268, −0.084]

Note: *p*-values that are bolded indicate statistically significant results.

**Table 3 behavsci-15-00899-t003:** The mean values (ms) and standard deviations of eye movement indicators within the three interest areas in Experiment 2.

		FFD		GD		SRT		TRT		RPR	
	Condition	*M*	*SD*	*M*	*SD*	*M*	*SD*	*M*	*SD*	*M*	*SD*
NP1	C-C	209	66	281	154	217	269	490	295	394	277
	C-I	204	59	268	136	382	341	638	367	464	276
	I-C	212	66	261	124	263	271	511	294	401	236
	I-I	212	65	281	141	374	334	645	344	471	270
V	C-C	218	69	245	99	126	186	374	206	896	533
	C-I	217	70	251	107	229	241	491	264	1318	725
	I-C	214	65	248	110	171	226	441	269	1064	647
	I-I	222	73	251	110	204	240	461	259	1209	701
NP2	C-C	215	67	275	137	143	214	433	256	541	361
	C-I	223	69	317	167	228	248	561	298	769	449
	I-C	211	64	278	142	178	230	467	259	624	403
	I-I	214	66	266	121	207	241	488	274	727	474

Note: FFD = First Fixation Duration; GD = Gaze Duration; SRT = Second Reading Time; TRT = Total Reading Time; RPR = Regression Path Reading Time. Means (*M*) and standard deviations (*SD*) were calculated based on the raw data. All mean values are reported in milliseconds (ms).

**Table 4 behavsci-15-00899-t004:** The fixed effects of the linear mixed-effects model for various eye movement indicators in the interest area in Experiment 2.

		NP1					V					NP2				
		*b*	*SE*	*t*	*p*	*CI*	*b*	*SE*	*t*	*p*	*CI*	*b*	*SE*	*t*	*p*	*CI*
FFD	Intercept	2.301	0.009	252.730	<0.001	[2.283, 2.319]	2.315	0.009	270.046	<0.001	[2.297, 2.333]	2.314	0.008	291.626	<0.001	[2.298, 2.330]
	Stroop	0.010	0.007	1.322	0.186	[−0.004, 0.024]	0.002	0.009	1.189	0.852	[−0.016, 0.020]	−0.014	0.008	−1.623	0.113	[−0.030, 0.002]
	Sentence	−0.004	0.008	−0.564	0.577	[−0.020, 0.012]	0.005	0.008	0.601	0.551	[−0.011, 0.021]	0.010	0.008	1.214	0.232	[−0.006, 0.026]
	Stroop:Sentence	0.006	0.017	0.339	0.737	[−0.027, −0.039]	0.016	0.017	0.903	0.376	[−0.017, 0.049]	−0.010	0.015	−0.664	0.511	[−0.039, 0.019]
GD	Intercept	2.386	0.015	156.654	<0.001	[2.357, 2.415]	2.358	0.012	204.259	<0.001	[2.334, 2.382]	2.404	0.015	155.424	<0.001	[2.375, 2.433]
	Stroop	−0.001	0.012	−0.058	0.954	[−0.025, 0.023]	0.001	0.010	0.043	0.966	[−0.019, 0.021]	−0.030	0.011	−2.634	**0.012**	[−0.052, −0.008]
	Sentence	0.006	0.012	0.049	0.625	[−0.018, 0.030]	0.006	0.010	0.593	0.557	[−0.014, 0.026]	0.022	0.011	1.951	0.059	[0.000, 0.044]
	Stroop:Sentence	0.045	0.029	1.577	0.123	[−0.012, 0.102]	−0.004	0.021	−0.177	0.861	[−0.045, 0.037]	−0.062	0.027	−2.285	**0.028**	[−0.115, −0.009]
SRT	Intercept	1.772	0.083	21.101	<0.001	[1.609, 1.935]	1.282	0.084	15.227	<0.001	[1.117, 1.447]	1.335	0.077	17.397	<0.001	[1.184, 1.486]
	Stroop	0.153	0.066	2.306	**0.028**	[0.024, 0.282]	0.030	0.084	0.354	0.726	[−0.135, 0.195]	0.062	0.088	0.705	0.486	[−0.110, 0.234]
	Sentence	−0.312	0.067	6.992	**<0.001**	[−0.443, −0.181]	0.411	0.104	3.936	**<0.001**	[0.207, 0.615]	0.313	0.086	3.653	**<0.01**	[0.144, 0.482]
	Stroop:Sentence	−0.448	0.211	−1.476	0.148	[−0.862, −0.034]	−0.365	0.196	−1.863	**0.07**	[−0.749, 0.019]	−0.331	0.162	−2.049	**0.048**	[−0.649, −0.013]
TRT	Intercept	2.680	0.021	125.543	<0.001	[2.639, 2.721]	2.571	0.020	130.994	<0.001	[2.532, 2.610]	2.617	0.019	138.227	<0.001	[2.580, 2.654]
	Stroop	0.021	0.017	1.243	0.222	[−0.012, 0.054]	0.015	0.016	0.936	0.356	[−0.016, 0.046]	−0.013	0.016	−0.785	0.437	[−0.044, 0.018]
	Sentence	0.109	0.017	6.513	**<0.001**	[0.076, 0.142]	0.073	0.021	3.535	**0.001**	[0.032, 0.114]	0.070	0.017	4.128	**<0.001**	[0.037, 0.103]
	Stroop:Sentence	−0.112	0.051	−0.226	0.822	[−0.212, −0.012]	−0.087	0.036	−2.405	**0.021**	[−0.158, −0.016]	−0.103	0.036	−2.884	**<0.01**	[−0.174, −0.032]
RPR	Intercept	2.558	0.023	112.614	<0.001	[2.513, 2.603]	2.960	0.026	115.300	<0.001	[2.909, 3.011]	2.728	0.022	124.387	<0.001	[2.685, 2.771]
	Stroop	0.008	0.017	0.486	0.631	[−0.025, 0.041]	0.016	0.018	0.895	0.367	[−0.019, 0.051]	0.011	0.017	0.651	0.520	[−0.022, 0.044]
	Sentence	0.061	0.016	3.834	**<0.001**	[0.030, 0.092]	0.121	0.022	5.551	**<0.001**	[0.078, 0.164]	0.111	0.018	6.124	**<0.001**	[0.076, 0.146]
	Stroop:Sentence	0.014	0.049	0.284	0.778	[−0.082, 0.110]	−0.119	0.044	−2.671	**0.011**	[−0.205, −0.033]	−0.108	0.047	−2.305	**0.027**	[−0.200, −0.016]

Note: *p*-values that are bolded indicate statistically significant results.

**Table 5 behavsci-15-00899-t005:** Results of Cross-Experiment Comparisons.

		NP1			V			NP2		
		χ^2^	*df*	*p*	χ^2^	*df*	*p*	χ^2^	*df*	*p*
FFD	str	2.941	1	0.086	0.910	1	0.340	1.372	1	0.241
	sen	0.769	1	0.380	0.502	1	0.478	1.732	1	0.188
	exp	0.194	1	0.659	1.545	1	0.213	0.447	1	0.503
	str:sen	0.143	1	0.705	0.521	1	0.470	0.265	1	0.606
	str:exp	0.526	1	0.468	0.509	1	0.475	1.678	1	0.195
	sen:exp	1.215	1	0.270	0.001	1	0.888	0.145	1	0.708
	str:sent:exp	2.452	1	0.117	0.211	1	0.645	0.140	1	0.707
GD	str	0.396	1	0.528	0.001	1	0.965	5.665	1	**0.017**
	sen	8.636	1	**0.003**	0.230	1	0.631	1.844	1	0.174
	exp	2.636	1	0.104	1.583	1	0.208	1.390	1	0.238
	str:sen	0.760	1	0.383	0.547	1	0.459	1.423	1	0.232
	str:exp	0.171	1	0.678	0.001	1	0.979	0.440	1	0.506
	sen:exp	5.932	1	**0.015**	0.123	1	0.725	3.928	1	**0.047**
	str:sent:exp	3.990	1	**0.046**	0.327	1	0.566	3.305	1	0.069
SRT	str	5.208	1	0.022	0.976	1	0.323	0.015	1	0.899
	sen	45.232	1	**<0.001**	30.776	1	**<0.001**	31.511	1	**<0.001**
	exp	13.269	1	**<0.001**	12.882	1	**<0.001**	12.622	1	**<0.001**
	str:sen	4.007	1	**0.045**	7.742	1	**0.005**	15.536	1	**<0.001**
	str:exp	1.151	1	0.283	0.272	1	0.601	1.048	1	0.305
	sen:exp	8.557	1	**0.003**	0.748	1	0.386	0.007	1	0.929
	str:sent:exp	0.403	1	0.525	1.350	1	0.245	4.353	1	**0.037**
TRT	str	3.390	1	0.065	0.542	1	0.461	1.001	1	0.317
	sen	41.959	1	**<0.001**	20.615	1	**<0.001**	29.673	1	**<0.001**
	exp	3.692	1	0.054	5.466	1	**0.019**	3.483	1	0.061
	str:sen	1.333	1	0.248	8.673	1	**0.003**	16.717	1	**<0.001**
	str:exp	0.179	1	0.671	0.773	1	0.379	0.003	1	0.956
	sen:exp	2.918	1	0.087	1.037	1	0.308	0.129	1	0.719
	str:sent:exp	4.259	1	**0.039**	0.471	1	0.492	0.384	1	0.535
RPR	str	2.236	1	0.134	0.275	1	0.599	0.070	1	0.789
	sen	15.790	1	**<0.001**	43.805	1	**<0.001**	61.035	1	**<0.001**
	exp	0.649	1	0.420	4.566	1	**0.032**	6.346	1	0.011
	str:sen	0.253	1	0.614	11.280	1	**<0.001**	12.109	1	**<0.001**
	str:exp	1.513	1	0.218	1.117	1	0.290	1.746	1	0.186
	sen:exp	1.247	1	0.264	2.971	1	0.087	1.314	1	0.251
	str:sent:exp	3.209	1	0.073	1.107	1	0.292	2.100	1	0.147

Note: FFD = First Fixation Duration; GD = Gaze Duration; SRT = Second Reading Time; TRT = Total Reading Time; RPR = Regression Path Reading Time; str = stroop; sen = sentence; exp = experiment; *p*-values that are bolded indicate statistically significant results.

## Data Availability

The datasets generated during and/or analyzed during the current study are available from the first author on reasonable request.

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
