# Peer review of "Transferable Modulation of Cognitive Control: The Cross-Task Role of Conflict Adaptation in Thematic Roles Assignment in Chinese"

_behavsci, 2025, doi:10.3390/bs15070899_

Round 1
Reviewer 1 Report
Comments and Suggestions for Authors
The authors investigated whether conflict adaptation elicited in general cognitive domains facilitates the resolution of conflicts arising from thematic role assignment during sentence comprehension. I recognize the potential of this manuscript for acceptance and publication, as it offers meaningful theoretical and practical contributions to our understanding of language processing. However, prior to publication, the authors need to clarify several issues:
Major comments:
Whether eye-tracking techniques, which offer higher temporal resolution, have resolved the issue of non-replicable cross-task conflict adaptation (Dudschig, 2022; Kan et al., 2013), and the reasons why this issue has or has not been addressed.
The authors should reorganize their rationale for conducting the current study. In my view, their innovation primarily lies in two aspects: first, their focus on adaptation triggered by response conflict rather than representational conflict; second, their use of eye-tracking techniques during visual reading instead of auditory comprehension. The authors should clearly separate these two points into distinct paragraphs, explicitly stating their research questions and justifications, rather than mixing them together.
In section 2.4, the sentence “Additionally, 20 questions are randomly selected from the pool of all sentences” may lead to ambiguity. If the intended meaning is that “a total of 20 questions were randomly distributed across the experiment and were based on the specific sentences currently being presented,” it might be clearer to revise the sentence accordingly.
Additionally, I have a few minor comments:
Regarding citations, at Line 136, the authors mention that "some researchers have argued that these studies predominantly rely on behavioral experimental methods." Precise references need to be provided here.
There is a problem with the layout of Table 2, Table 3, and Table 4. I am not sure if this is caused by the file conversion. Please check carefully.
I am not sure if the figure legend needs two spaces on the first line (in all figures). Please check carefully
In Figures 2 and 3, please clarify what ** and *** represent.
The authors need to rephrase their expressions in the Introduction, as there is an excessive use of phrases like "building on/upon...". Greater variety in wording would help improve readability and clarify the text's structure.
Reviewer 2 Report
Comments and Suggestions for Authors
Luo et al. use the conflict adaptation paradigm from Stroop to Sentence to investigate cross-task conflict adaptation. Prior literature shows discrepant evidence for cross-task conflict adaptation. Studies using the visual world paradigm find positive support and many self-paced reading studies find negative support. The current eye-tracking while reading paradigm found support of transfer from Stroop to (Im)plausible sentence reading in Chinese. In Experiment 1, late measures found incongruent stroop facilitated implausible sentence reading and slowed down plausible sentence reading. In Experiment 2, where Stroop conflict is argued to be present at both representational and response levels, similar results were found for the plausible sentences, but the effects for implausible sentences was found earlier – Gaze Duration. The results are interpreted in terms of the conflict monitoring account. However, the design and results don’t fully justify these interpretations. Further analyses would be helpful to interpreting the results. I elaborate on these points below as well as concerns about how cognitive control applies in these implausible sentences.
Conflict and Resolution in Implausible Sentences:
“Furthermore, a significant conflict adaptation effect was observed in late-stage indicators within the interest area, suggesting that exposure to conflict in a preceding trial enhances the engagement of cognitive control resources, thereby improving processing efficiency for subsequent conflicting information.”
I think it is critical to explain how conflict monitoring applies to implausible sentences. Prior work has made a clear case for how a conflict monitoring account of cognitive control can apply to garden paths, but it is not transparent how cognitive control operates in the case of implausible sentences. What conflict is being identified and how does resolution apply and when in the sentence are these two things occurring? How is the reduced regression-path time at the object of implausible sentences following incongruent stroop interpreted in terms of sentence processing. Is it facilitating sentence processing in some way – thematic role assignment - or is it just reducing time spent being surprised by unexpected and implausible information?
Regions Analysed:
The authors report results at a single region. Please provide clear motivation for this being the critical region. Additional regions should be analysed to better interpret the results. I would suggest looking at the subject and verb regions as well. You wouldn’t expect to see any effects at the subject region. At the verb region, it seems there is already an implausibility (“the tires checks”). Do you see the same effect at the verb where the lexical item is identical across congruent and incongruent sentences (unlike at the object - the critical region)?
Plausible Sentence Effects:
If these first two points can be satisfactorily addressed, then there remains the question of why there are effects in the congruent sentence condition? This result is not predicted by a conflict monitoring account of cognitive control, as there is no conflict in the congruent sentence.
“Specifically, after encountering incongruent Stroop trials, reading times for subsequent congruent sentences are significantly prolonged, reflecting a sustained sensitivity to previously encountered conflicts.”
How can these results be interpreted? They are not consistent with conflict monitoring that selectively expects facilitation in “incongruent” sentence reading and no effect with “congruent” sentences where there is no conflict. Likewise the prior literature cited (Hsu & Novick, 2016; Hsu et al., 2020) did not see effects on the congruent sentence. Some discussion of this unexpected result is required. Do you expect to see this result throughout the sentence or only at the “region of interest”? Again, analysing further regions would help in understanding this data point.
Representational conflict vs Response Conflict:
Experiment 1 also includes response conflict since the colour words in the incongruent condition correspond to button responses. So there is conflict in terms of the button to press – either the one corresponding to the colour word or the one corresponding to font colour. It is not the case that there is ONLY representational conflict. This leads to the question of what is responsible for the different effects across experiments (next point).
Effects across Experiments:
I’m not entirely clear why some results from Experiment 1 are replicated and others are not. Specifically, incongruent Stroop reduces Gaze duration measures at critical region in implausible sentences in Experiment 2, whereas later effects were observed in Experiment 1 (regressions and regression path). Why would that be? The specific version of response conflict changes expectations? That seems hard to explain.
I’d also suggest running an analysis where “Experiment” is a between subject factor to ensure any differences across experiments are actually significant.
Trial Order:
A pseudorandom order is mentioned. Is the reason it is not completely random because of the “cycle” structure? Or were there other principles applied to the pseudorandomisation.
Was the pseudorandom order the same across participants assigned to a list?
Was there an cycle order included in the model to eliminate any order effects?
Minor Concerns:
“By introducing response conflict, Liu et al. (2012) also observed a robust conflict adaptation effect: conflict trials preceded by a conflict trial elicited significantly faster reaction times compared to those following congruent trials.”
But there is also representational conflict so how can you ensure the effect is due to response conflict?
Reviewer 3 Report
Comments and Suggestions for Authors
This manuscript examines whether conflict adaptation elicited by non-linguistic tasks can influence sentence processing, specifically through thematic role assignment in Chinese. Across two well-controlled experiments using Stroop paradigms and eye-tracking measures, the authors demonstrate that both representational and response conflict modulate syntactic processing, with response conflict producing earlier and stronger effects. The research is methodologically sound, and the findings are clear and well-reported.
The study makes a valuable contribution to the literature on cognitive control and language processing, particularly by showing that control mechanisms can operate across task boundaries. The use of Chinese adds cross-linguistic relevance, and the application of eye-tracking allows for fine-grained analysis of processing dynamics. The experimental design is appropriate, and the statistical modeling is rigorous.
However, the manuscript would benefit from several improvements. First, the literature review, although generally appropriate, relies heavily on earlier sources. Some key topics—such as the dynamics of cognitive control, cross-task transfer, and sentence parsing—would be better supported with a more updated review. While foundational works are necessary, the review should more clearly reflect the current state of the field.
Second, the discussion could be developed further. The interpretation of the timing differences between conflict types is plausible but underexplored. A deeper reflection on the mechanisms involved in early versus late adaptation would enhance the theoretical impact. The broader implications of the findings, such as their relevance to other languages or applied settings, are not fully addressed. Also, the nature of the linguistic conflict requires clarification, particularly regarding whether the incongruent sentences involved syntactic ambiguity, semantic implausibility, or both—an issue that affects how the adaptation effects are interpreted.
Finally, the presentation of results in the tables would benefit from improvement. Currently, key terms such as FFD, GD, TRT, and RPR are not explained, nor is it clear whether the values shown are raw or transformed. Including short footnotes or captions to clarify abbreviations, units, and transformations would make the tables more accessible and transparent for readers unfamiliar with these measures.
In sum, this is a strong and original study that adds important evidence to the understanding of how cognitive control interacts with language processing. I recommend acceptance with minor revisions, focused on updating the theoretical framing, clarifying interpretations, and improving the presentation of results.
Round 2
Reviewer 2 Report
Comments and Suggestions for Authors
The revisions, including additional analyses, make for a much improved manuscript. I think it is getting closer to publication. However, the additional results also raise some concerns and the explanation of how conflict resolution applies to implausible sentences is incomplete.
- Resolution of unacceptable sentences. The revisions help provide some further explanation but I think they are incomplete as they stand.
“Resolving such conflicts requires substantial cognitive control…to achieve a coherent interpretation.”
“At this point, cognitive control mechanisms are likely activated to suppress inappropriate prepotent interpretations and facilitate the reanalysis required for accurate thematic role assignment”.
I don’t know what assumptions are being made, but there is no coherent interpretation of the “the apple ate the boy”. It is an example of an incoherent sentence. Indeed your acceptability judgments support this, as the “incongruent” sentences were judged as unacceptable. The current sentences differ from Thotharithi where plausibility was lower but not impossible. Rabbits can chase foxes, albeit the probability is higher for foxes to chase rabbits. Unlike garden path sentences I’m not sure there is much work exploring the comprehension of implausible sentences to make clear which initial interpretation comprehenders make and whether there is evidence of reanalysis. Please make explicit, if it is possible, what are the initial “prepotent” and final interpretations. If there isn’t further supporting data for this, this should also be acknowledged as a limitation in the General Discussion.
- Sentence Effect at NP1. In both Experiment 1 and 2, there is an effect of sentence condition before the condition should be evident, at NP1. Congruent sentences have shorter gaze/RPR duration, but congruency cannot be determined at NP1. This seems to suggest a confound in your manipulation, which poses concerns for the other regions downstream.
- Marginal Effects. In inferential statistics, results are either significant or not. Please remove mention of “marginal effects” in figures and text.
- Between Experiment Results. Report the full set of results (e.g. in a Table) when collapsing across experiments to understand which are reliable across your Stroop manipulation. Conflict monitoring as a theory does not predict a difference across the two Stroop tasks, so any differences need to be acknowledged as post-hoc interpretations.
- Counterbalancing. To counterbalance order, you’d need 4 lists. Presumably what changes across your list is whether the sentence is “congruent” or “incongruent” but then that means you have a different order of conditions across your lists and they aren’t completely counterbalanced. If in List 1 you have the first 4 trials as CI, IC, II, CC… then in List 2 you have CC, II, IC, CI but that’s not completely counterbalanced.
- Figure 2: some confidence intervals are missing
Round 3
Reviewer 2 Report
Comments and Suggestions for Authors
Thank you for those responses. They help with making the manuscript more precise and clear. I still find the application of conflict monitoring to implausible sentences to lack precision and is a bit vaguely explained in terms of correcting thematic role assignment. It remains unclear what the initial assignment is. If that could be made explicit, I think the argument would be much stronger and more convincing. Nonetheless, I think the results show that a preceding Stroop facilitates the processing of implausible sentences. I'll leave it to further work to better understand why that is.
